# MatRL: Provably Generalizable Iterative Algorithm Discovery via Monte-Carlo Tree Search

## Abstract

Iterative methods for computing matrix functions have been extensively studied and their convergence speed can be significantly improved with the right tuning of parameters and by mixing different iteration types (Higham & Schreiber, 1990). Hand-tuning the design options for optimal performance can be cumbersome, especially in modern computing environments: numerous different classical iterations and their variants exist, each with non-trivial per-step cost and tuning parameters. To this end, we propose **MatRL** – a reinforcement learning based framework that automatically discovers iterative algorithms for computing matrix functions. The key idea is to treat algorithm design as a sequential decision-making process. Monte-Carlo tree search is then used to plan a *hybrid* sequence of matrix iterations and step sizes, tailored to a specific input matrix distribution and computing environment. We also show that the learned algorithms provably generalize to sufficiently large matrices drawn from the same distribution. Finally, we corroborate our theoretical results with numerical experiments demonstrating that MatRL produces algorithms that outperform various baselines in the literature.

## 1 Introduction

Matrix functions are everywhere - ranging from classical applications in control theory (Denman & Beavers Jr, 1976), high-dimensional ODEs (Higham, 2008), theoretical particle physics (Chen & Chow, 2014), (Lin et al., 2009), Markov models (Waugh & Abel, 1967), to some recent applications in machine learning, e.g. covariance pooling (Li et al., 2018), (Wang et al., 2020), (Song et al., 2021), graph signal processing (Defferrard et al., 2016), (Maskey et al., 2023), contrastive learning (Richemond et al., 2023), and optimizer design (Gupta et al., 2018), (Jordan et al., 2024b), (Ahn & Xu, 2025). For more applications see (Higham, 2008) and references therein. It isn't a surprise that computing matrix functions in a fast, precise, and stable manner has received numerous attention and many have tried to develop better algorithms with guarantees that it will work in some sense.

Iterative algorithms to compute matrix functions are particularly attractive in modern applications as they can avoid computing the matrix function directly using the singular value decomposition (SVD). Further, termination criteria can be chosen based on the needs of the application, which can lead to faster and more stable algorithms. Additionally, these algorithms are differentiable, leading to potential applications in auto-differentiation based settings (Song et al., 2022). From an algorithm design perspective, it would be ideal to find an algorithm that is computationally efficient, numerically stable and has faster convergence guarantees. One can imagine using different iterations from different algorithms at each step (e.g. mixing the algorithms) and specifically tuning the parameters jointly to accelerate the algorithm. However, due to the large search space (see Section 2.1 and references therein to see a vast array of existing iterative algorithms) and highly sensitive and non-trivial per iteration costs in modern computing environments, handcrafting the ideal algorithm is tedious and impractical.

A motivating experiment is presented in Table 1. In single precision, the cost of computing a $4096 \times 4096$ inverse is roughly five times that of a matrix multiply on both CPU and GPU. However, switching to double precision more than halves this ratio – dropping to about 2.9x on CPU and 1.7x on GPU – because matrix multiplies become comparatively more expensive. This pronounced

Table 1: Computation times for matrix inverse and multiplication ($4096 \times 4096$) on CPU and GPU in single (float32) and double (float64) precision. The right-most column shows the ratio of inverse time to matmul time. All computations done using PyTorch on an RTX A6000 GPU.

| Device | Precision | Inverse (ms) | MatMul (ms) | Inv : MatMul ratio |
|--------|-----------|--------------|-------------|--------------------|
| CPU    | float32   | $251.2 \pm 8.7$  | $48.8 \pm 1.8$   | 5.15 |
|        | float64   | $356.6 \pm 16.3$ | $122.9 \pm 4.6$  | 2.90 |
| GPU    | float32   | $28.3 \pm 0.5$   | $5.6 \pm 0.1$    | 5.05 |
|        | float64   | $423.9 \pm 6.9$  | $252.6 \pm 6.4$  | 1.68 |

change in relative costs with precision underscores the need for automatic algorithm discovery, as the optimal sequence of matrix iterations will depend sensitively on both hardware and numerical precision. Besides the change in relative speeds, the gains from switching to GPU turns into a slow-down with double precision for both inverse and matmuls. This is due to the significantly lower emphasis on high precision compute capability in modern GPUs.

We propose an automated method based on Monte-Carlo tree search to decide which combination of iterations and parameters one should use given a desiderate from the user. Our solution assumes that the matrix of interest is sampled repeatedly from a certain random matrix distribution, and we want to find a good algorithm for that matrix distribution. The main idea is that iterative algorithms can be understood as a sequential decision-making process: at each step, one should choose which iterative method to use and which parameters to use in that iterative method. This corresponds to choosing actions in decision making, where the choice leads to the next step. At each step we get a certain reward signal based on the given desiderata, so that we can evaluate whether the action was worth it or not. Finding a good algorithm mounts to finding a good planning strategy for the given environment. Specifically,

- We propose **MatRL** (Algorithm 7), an automated algorithm searching scheme based on Monte-Carlo tree search. In line with recent algorithm-discovery frameworks (Fawzi et al., 2022), (Mankowitz et al., 2023), (Collet et al., 2025), we define a novel search space of algorithms and propose an efficient strategy to explore it.

- The algorithms we find are faster than existing baselines, and even faster than implementations in torch.linalg. Moreover, the found algorithms differ with problem sizes, computation environment, and precision (Section 5.3), meaning that MatRL adapts to different enviroments with ease.

- We have a guarantee using random matrix theory that the found algorithm will generalize to different matrices drawn from the same distribution (Section 4), and matrices drawn from the distribution with identical limiting eigenvalues.

- We additionally explore extensions to rectangular matrices and out-of-distribution generalization, demonstrating an application of the approach through the computation of matrix sign function in NanoGPT training (Section 5.2

The paper is organized as the following: in Section 2 we discuss relevant background on iterative matrix function computation algorithms, Monte-Carlo tree search and learning algorithms via RL. Next we describe our environment in Section 3 by describing how the states, actions, state transition, and reward signals are defined. In Section 4, we provide the generalization guarantee stemming from limiting distribution of the spectrum. We show experimental results in Section 5 showing the performance and adaptivity of MatRL, as well as its application to rectangular matrices and out-of-distribution generalization. We conclude the paper in Section 6.

## 2 BACKGROUND

For readers unfamiliar with Monte-Carlo tree search, we provide a brief overview of the concept as well as general applications in Section C.1.

## 2.1 ITERATIVE METHODS TO COMPUTE MATRIX FUNCTIONS

One of the basic ideas of obtaining an iteration to compute matrix functions is using Newton's method (Higham, 2008). For instance, say we want to compute the matrix square root. We would like to compute the root of the function $f(X) = X^2 - A$, hence Newton's method can be written as

$$X_{k+1} = X_k - f'(X_k)^{-1}f(X_k) = X_k - \frac{1}{2}X_k^{-1}(X_k^2 - A) = \frac{1}{2}(X_k + X_k^{-1}A). \tag{1}$$

Newton's method uses a first-order approximation of $f$ at $X_k$: we may use higher order approximations to obtain Chebyshev method (Li et al., 2011) or Halley's method (Nakatsukasa et al., 2010), (Guo, 2010). These higher-order methods converge cubically with appropriate initialization, but each iteration is slower than Newton's method. We could also use inverse-free methods such as Newton-Schulz and its variants (Higham, 2008), (Higham, 1997), which approximates $X_k^{-1}$ with a polynomial of $X_k$.

Appropriate scaling and shifting of the spectrum to yield faster convergence has been a popular idea (Nakatsukasa et al., 2010), (Byers & Xu, 2008), (Byers, 1987), (Chen & Chow, 2014), (Iannazzo, 2003), (Hoskins & Walton, 1979), (Pan & Schreiber, 1991). The intuition is that by introducing additional parameters and solving an optimization problem on the spectrum, we can find a sequence of optimized parameters that depend on the spectrum of $A$, the matrix that we would like to compute matrix function. For instance, (Chen & Chow, 2014) finds a cubic function $h : [0,1] \to [0,1]$ that maximizes $h'(0)$ to find better scaling of Newton-Schulz iteration. One drawback of these approaches is that in many cases we need to compute smallest / largest eigenvalues of the matrix (Byers & Xu, 2008), (Chen & Chow, 2014), (Pan & Schreiber, 1991), (Nakatsukasa et al., 2010), (Hoskins & Walton, 1979), which may be expensive to compute. In this case we use approximations of the smallest eigenvalue, such as $1/\|A^{-1}\|_F$. Another drawback is that each new scaling scheme needs a complicated mathematical derivation and proof.

Naively applying Newton's method can be unstable for computing matrix square-root and $p$-th root. To ensure stability, we can introduce an auxillary variable $Y_k$ and simultaneously update $X_k$ and $Y_k$ (Higham, 1997), (Iannazzo, 2006). We will refer to such iterations as coupled iterations. Coupled iterations are obtained by manipulating the formula so that we do not have $A$ in the iteration. Going back to Newton's method for computing square roots in Eq. (1), we can introduce an auxillary variable $Y_k = A^{-1}X_k$ and initialize $X_0 = A, Y_0 = I$ to obtain the coupled iteration known as Denman-Beavers iteration (Denman & Beavers Jr, 1976),

$$\begin{cases} X_{k+1} = \frac{1}{2}(X_k + Y_k^{-1}) \\ Y_{k+1} = \frac{1}{2}(Y_k + X_k^{-1}). \end{cases} \tag{2}$$

Using perturbation analysis, (Higham, 1997) shows that the iteration in Eq. (1) is unstable when the condition number $\kappa(A) > 9$, whereas the iteration in Eq. (2) is stable.

Iterative algorithms are not limited to Newton's method. We may have fixed-point iterations such as Visser iteration (Higham, 2008), where we iteratively compute $X_{k+1} = X_k + \alpha_k(A - X_k^2)$ to compute matrix square root. We may also use higher-order rational approximations of the function of interest to obtain algorithms that converge in only a few steps (Nakatsukasa & Freund, 2016), (Gawlik, 2019), and with sufficient parallelization they can be faster than existing methods.

## 2.2 AUTOMATED ALGORITHM DISCOVERY

We are not the first to discover algorithms using ideas from sequential decision-making. RL-based approaches which parametrize the policy as a neural network have proven to be successful: (Li & Malik, 2017) learns to optimize neural networks using an RL framework. In their framework, the states consist of past variables $x_i$, past gradients $\nabla f(x_i)$, and past objectives $f(x_i)$, and the policy aims to learn appropriate $\Delta x$. (Fawzi et al., 2022) used reinforcement learning to find faster matrix multiplication algorithms, (Mankowitz et al., 2023) finds faster sorting algorithms, and (Collet et al., 2025) uses evolutionary algorithm to find better compression algorithms given a format of the data. (Khodak et al., 2024) has a similar flavor with our work, where they use contextual bandits to optimize relaxation parameters in symmetric success-over-relaxation.

## 3 MATRL: ITERATIVE MATRIX FUNCTION ALGORITHM SEARCH VIA RL

### 3.1 OBJECTIVE

Let $A \sim \mathcal{D}$, where $\mathcal{D}$ is a distribution over symmetric random matrices defined in $\mathbb{R}^{n \times n}$ and $f : \mathbb{R} \to \mathbb{R}$ is the function that we would like to compute. We follow the definition of matrix functions in (Higham, 2008). For a symmetric matrix $A$, when we diagonalize $A = UDU^T$ for an orthogonal matrix $U$, $f(A) = Uf(D)U^T$ where $f(D)$ applies $f$ to the diagonal entries of $D$. Denote $f_1(X, Y, A, a_1), f_2(X, Y, A, a_2), \cdots f_m(X, Y, A, a_m)$ as $m$ choices of iterations that we can use, $a_j \in \mathbb{R}^{n_j}$ as tunable parameters for each $f_j$, and $T_j$ the wall-clock time needed to run $f_j$. We use two variables $(X, Y)$ as input to accomodate coupled iterations, and $Y$ is not used for iterations that are not coupled. For instance, for $f = \sqrt{\cdot}$, $f_1(X, Y, a_1)$ can be the scaled Denman-Beavers iteration

$$f_1(X, Y, A, a_1) = \Big( \frac{1}{2}(a_{11}X + (a_{12}Y)^{-1}), \frac{1}{2}(a_{12}Y + (a_{11}X)^{-1}) \Big),$$

whereas $f_2(X, Y, A, a_2)$ can be the scaled Visser iteration

$$f_2(X, Y, a_2) = \Big( a_{21}X + a_{22}(A - X^2), Y \Big). \tag{3}$$

Here, $n_j$ denotes the number of tunable parameters in iteration $f_j$. Also, assume the error tolerance $\epsilon_{\text{tol}}$ is given.

Now, we specify the ***class*** of matrix iterations and the custom loss function $\mathcal{L} : \mathbb{R}^{n \times n} \times \mathbb{R}^{n \times n} \to \mathbb{R}$ that determines the termination condition. For this we define congruence invariant matrix functions. Note that the actual loss function and the matrix iterations should later be specified by the user.

**Definition 1.** *(Congruence Invariant Diagonal Preserving) Let* $f : (\mathbb{R}^{n \times n})^k \to \mathbb{R}^{m \times m}$ *a matrix function that takes $k$ matrices as input and outputs a matrix. If* $f(QX_1Q^T, QX_2Q^T, \cdots QX_kQ^T) = Qf(X_1, X_2, \cdots X_k)Q^T$ *for all orthogonal $Q$, $f$ is congruent invariant. If* $f(X_1, X_2, \cdots, X_k)$ *is diagonal for diagonal $X_1, X_2, \cdots X_k$, $f$ is diagonal preserving. For functions that take matrix tuple as an input and outputs a matrix tuple, $F : (\mathbb{R}^{n \times n})^k \to (\mathbb{R}^{m \times m})^l$, we call $F$ congruent invariant diagonal preserving if $F = (f_1, f_2, \cdots f_l)$ and all $f_i$ are congruent invariant and diagonal preserving for $i \in [l]$.*

We limit the matrix iterations and the loss function to be congruent invariant diagonal preserving functions, regarding $A$ also as an input. For instance, the Newton-Schulz iteration to compute matrix inverse (Pan & Schreiber, 1991), $f(X, A) = 2X - XAX$ is a congruence invariant diagonal preserving function and the iteration falls into our framework. Such limitation will enable us to understand actions and losses as functions of the spectrum of $X, Y$, and $A$, and as we will see in the next section, it will enable us to see the whole environment only as a function of the spectrum.

Our objective is, given a sample $A$ from $\mathcal{D}$, find a sequence of iterations and coefficients $f_{t_1}(\cdot, a_{t_1}), f_{t_2}(\cdot, a_{t_2}), \cdots f_{t_N}(\cdot, a_{t_N})$ that is a solution to

$$\max_{N, t_i \in [m], a_{t_i} \in \mathbb{R}^{n_{t_i}}, \, i \in [N]} \quad -\sum_{i=1}^{N} T_{t_i} \quad \text{subject to} \quad \mathcal{L}(X_{N+1}, A) \leq \epsilon_{\text{tol}}, \tag{4}$$

where $X_0 = A, Y_0 = I$ or $X_0 = I, Y_0 = A$ depending on $f$, $(X_{k+1}, Y_{k+1}) = f_{t_k}(X_k, Y_k, a_{t_k})$, $\mathcal{L}$ is a loss function to be specified later, and $N$ is also optimized. Note that similar ideas have also appeared in optimal control (Evans, 1983) in the context of minimum-time control problems.

The optimal solution of Eq. (4) naturally corresponds to finding the optimal iterative algorithm

$$X_{k+1}, Y_{k+1} \leftarrow f_{t_k}(X_k, Y_k, a_{t_k}), \quad k \in [N].$$

that arrives as $\mathcal{L}(X, A) \leq \epsilon_{\text{tol}}$ as fast as it can.

### 3.2 THE ENVIRONMENT

Here we elaborate how we formulate the problem in Eq. (4) to a sequential decision-making problem by describing $\mathcal{E} = (\mathcal{S}, \mathcal{A}, \mathcal{T}, r, t)$, which is a 5-tuple that describes the environment. $\mathcal{S}$ denotes the set of states, $\mathcal{A}$ denote the set of actions that one can take in each state, $\mathcal{T} : \mathcal{S} \times \mathcal{A} \to \mathcal{S}$ gives how

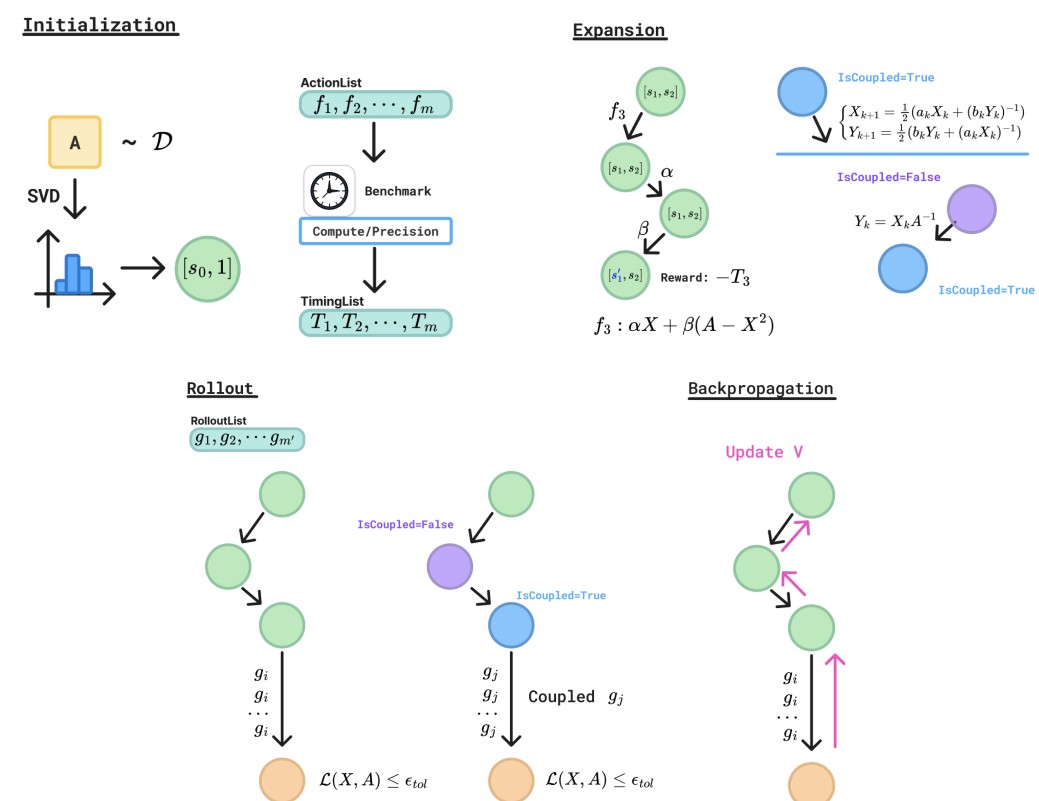

Figure 1: A schematic that summarizes details of MatRL at each stage of Monte-Carlo tree search. At initialization, we initialize the root node with the spectrum of $A$ and precompute the time needed for each action. At expansion stage, we choose action/each parameters separately, and expand with coupling actions only when the state is coupled. In the rollout stage we use prespecified rollout list which are already working baselines. If we want to rollout with coupled iteration but the state is not coupled, we use the coupling action. At backpropagation stage we update the value of each node.

state transition occurs from state $s$ when we apply action $a$, $r : \mathcal{S} \times \mathcal{A} \to \mathbb{R}$ gives the amount of reward one gets when we do action $a$ at state $s$, and $t : \mathcal{S} \to \{0, 1\}$ denotes whether the state is terminal or not.

The simplest way to define the state and action variables is by setting each state as a tuple $(X, Y)$ that corresponds to the matrices $(X_k, Y_k)$, and setting each action as a $n_j+1$ - tuple $(j, a_{j1}, a_{j2}, \cdots a_{jn_j})$, where $j \in [m]$ denotes the iteration $f_j$ and $a_{j1}, a_{j2}, \cdots a_{jn_j}$ denotes the parameters for $f_j$. The state transition $\mathcal{T}$ simply becomes

$$\mathcal{T}(X, Y, j, a_{j1}, a_{j2}, \cdots a_{jn_j}) = f_j(X, Y, a_j).$$

At each transition, we get rewarded by $-T_j$, the negative time needed to run the iteration. The terminal state is when $\mathcal{L}(X, A) \leq \epsilon_{\text{tol}}$. Our environment stems from this basic formulation, but we have important implementation details that we elaborate below.

**Spectrum as state variables** Imagine we follow the simple implementation above and use the actual matrices $(X, Y)$ while doing Monte-Carlo tree search(MCTS). Each MCTS run will correspond to running the matrix-function algorithm once, and as finding a good algorithm typically requires on the order of 1e5 MCTS runs, the time needed to find an algorithm can be huge.

Instead, we use $(s_1, s_2)$, where $s_1, s_2 \in \mathbb{R}^n$ correspond to the eigenvalues of $X, Y$. We initialize $X_0, Y_0$ with the spectrum of $A$ only once. Such parametrization is justified by the fact that both the transition $f_k(X, Y, A, a) = (X', Y')$ and the loss $\mathcal{L}$ can be expressed only using the spectrum for our iterations of interest (see Section A for a proof). Now each state transition and loss function

computation becomes a function of the spectrum, not the function of a matrix, which reduces the computational complexity of each MCTS run from $O(n^3)$ to $O(n)$ and makes the search feasible.

For this implementation we have a one time cost to measure the time needed for each iteration, as there is no actual matrix computation in the searching stage. Given precision, compute environment, and matrix size, we run each iterative method multiple steps and log the average runtime needed for each iterative method. This becomes the penalty $T_j$ of using iteration $f_j$ for $j \in [m]$, and we reuse this value every time we use the iteration $f_j$ instead of performing the actual matrix computation.

**Decoupling actions** At each state we have $n_j + 1$ tuple of possible actions. If we directly apply MCTS search on the $n_j + 1$ - tuple, we need to visit exponentially many nodes in the expansion stage which makes the search far from being efficient. Rather, we decouple the state transition $\mathcal{T}$ into $n_j + 1$ stages: at stage $1, 2, \cdots n_j$, only the parameters $a_{j1}, \cdots a_{jn_j}$ are chosen. At state $n_j + 1$, we get rewarded by $-T_j$, choose next iteration, and state transition happens.

This implementation allows MCTS to focus on completing sets of "promising" parameter choices rather than exhaustively exploring all parameter tuples - enabling efficient search.

**Dealing with coupled iterations** For computing matrix roots, some iterations are coupled (e.g., Denman-Beavers), while others like the scaled Visser iteration are not, making it challenging to mix them directly. Applying uncoupled iterations alone can break essential relationships (e.g., $Y_k X_k^{-1} = A$ for Denman-Beavers), potentially leading to incorrect results. To address this, we propose either augmenting uncoupled steps with coupled ones or tracking a boolean flag IsCoupled that governs when and how to restore consistency between variables before proceeding. See Section C for a detailed explanation.

**Extending to rectangular matrices** We can extend the environment to rectangular matrices if we assume the following: first, given matrix $A = U\Sigma V^t$, we define the matrix function to be $f(A) = Uf(\Sigma)V^t$. Second, we let $X_0 = A$ and choose the actions to satisfy $X_k = Uf_k(\Sigma)V^t$. An example of such choice is the Newton-Schulz iteration in the Muon optimizer (Jordan et al., 2024b). At last, the loss only depends on the spectrum of $X_k$ and $A$. If all conditions satisfy, we can simply use the spectrum of $A$ as the state variable and use the same framework to discover iterative matrix function algorithms for rectangular matrices.

### 3.3 The Searching Strategy

Here we describe the details of Monte-Carlo tree search in the predescribed environment $\mathcal{E}$. Let's note $n, v : \mathcal{S} \to \mathbb{N}$ the number of child nodes and visit count of that node, respectively. If all parameters $j, a_{j1}, \cdots a_{jn_j}$ are chosen and the state is ready for state transition, we call the state *transitionable*. We use progressive widening (Couëtoux et al., 2011) to deal with the continuous parameter space: during the selection stage, if the node is transitionable and hasn't visited all possible children, or the number of child nodes $n(s) \leq Cv(s)^\alpha$ for some hyperparameters $C, \alpha$, we go to the expansion step. If not, we choose the next node with UCT (Kocsis & Szepesvári, 2006). In the expansion stage, we add a child node. Choosing the iteration is discrete and we choose them depending on the IsCoupled flag. To choose the parameters, we first sample randomly for $E$ steps, then jitter around the best parameter found. In the rollout stage, we have predetermined baseline algorithms and run one of them to estimate the value of the state. At last, we backpropagate by using the Bellman equation

$$V_s = \max_{a \in \mathcal{A}} V_{\mathcal{T}(s,a)} + r(s,a).$$

In Fig. 1 we summarize the details of the environment and the search strategy of MatRL. Details on the parameters for each experiment and a thorough description of MatRL can be found in Section C. Specifically, the set of actions and rolloutlist can be found in Section C.4, and the list of all actions and their parametrization can be found in Table 6 in Section C.

## 4 Random Matrices and Generalization Guarantees

The objective in Eq. (4) aims to find the optimal algorithm for a given matrix $A$ sampled from $\mathcal{D}$. Here we show that under certain assumptions, the found iterative algorithm has a sense of generalization capability to a different matrix distribution $\mathcal{D}'$ with the same limiting distribution, i.e. the distribution as $n \to \infty$. The assumption that two distributions having the same limiting

distribution is motivated from the celebrated Marchenko-Pastur theorem, which states that for a random matrix distribution $\mathcal{D}$, the empirical spectrum $\mu_n$ converges in distribution to some limiting distribution $\mu^*$ with probability 1 if $\mathcal{D}$ satisfies certain conditions (Yaskov, 2016).

Two main ideas for the proof is: first, the loss curve only depends on the spectrum. Second, if the limiting distributions are identical for $\mathcal{D}$ and $\mathcal{D}'$, the sampled matrices' spectrum will be similar if the matrix is sufficiently large. Hence, the loss curve will be similar for two matrices $X \sim \mathcal{D}$, $Y \sim \mathcal{D}'$ for sufficiently large matrices, and if the algorithm works well for $X$, it will work well for $Y$. The specific guarantee that we have is in Proposition 1, which states that for sufficiently large matrix size $m$, the $k$ - th step performance is similar for two different matrices sampled from each distribution. The proof is deferred to Section A.

**Proposition 1.** *(Generalization of the discovered algorithm) Say we have a sequence of symmetric random matrix distributions $\mathcal{P}_m, \mathcal{Q}_m$ defined in $\mathbb{R}^{m \times m}$, and denote random matrices sampled from $\mathcal{P}_m$, $\mathcal{Q}_m$ as $X, Y$. Let the empirical eigenvalue value distribution of $X \sim \mathcal{P}_m, Y \sim \mathcal{Q}_m$ be $\mu_m(X), \nu_m(Y)$, and their support be $S_m, S'_m$, respectively. Now, suppose*
*(i) (Identical limiting distribution)*

$$\mathbb{P}(\mu_m(X) \Rightarrow \mu^*) = \mathbb{P}(\nu_m(Y) \Rightarrow \mu^*) = 1,$$

*i.e. both $\mu_m(X)$ and $\nu_m(Y)$ converges weakly to a common distribution $\mu^*$ with probability 1.*
*(ii) (Interval support of the limiting distribution) The support of $\mu^*$ is an interval $[a, b]$.*
*(iii) (Convergence of support) We have*

$$\lim_{m \to \infty} \mathbb{P}(S_m \subseteq [a - \epsilon, b + \epsilon]) = \lim_{m \to \infty} \mathbb{P}(S'_m \subseteq [a - \epsilon, b + \epsilon]) = 1,$$

*for all $\epsilon > 0$.*
*With the assumptions above, let $f$ be the matrix function we would like to compute, $f_k^*$ the step $k$ transformation of eigenvalues of the algorithm found by MatRL (i.e. $X_k = Q f_k^*(\Sigma) Q^T$ for $A = Q \Sigma Q^T$), and $L : \mathbb{R} \times \mathbb{R} \to \mathbb{R}$ be the loss. Assume $f, f_k^*, L$ are continuous in $[a - \epsilon_0, b + \epsilon_0]$ for some $\epsilon_0 > 0$. Write the empirical loss of the random matrix $X$ as*

$$\mathcal{L}_k(X) = \frac{1}{m} \sum_{i=1}^{m} L(f(\lambda_i), f_k^*(\lambda_i)),$$

*where $\lambda_i$ are eigenvalues of $X$.*
*Then, there exists $M_{\epsilon, \delta}$ such that*

$$m \geq M_{\epsilon, \delta} \Rightarrow \mathbb{P}_{X \sim \mathcal{P}_m, Y \sim \mathcal{Q}_m}[|\mathcal{L}_k(X) - \mathcal{L}_k(Y)| < \epsilon] \geq 1 - \delta.$$

## 5 EXPERIMENTS

In this section we specify the experimental details and present the discovered algorithms and their performance. Due to space issues, we cannot record every single experiment that was done in the main paper, and the full experimental results can be checked in Section D. Here we only introduce the most relevant experiments.

For our experiments, we either use a synthetic random matrix (e.g. Wishart random matrices, random matrices with spectrum generated from sampling $Unif[-1, 1]$), or real-world matrix datasets (e.g. CIFAR-10, momentum states of Muon optimizer in NanoGPT training repository). For the latter, we explicitly separate the dataset into train and test datasets. For all experiments except for the NanoGPT experiment, we train five different algorithms and test with 10 randomly sampled test matrices. For NanoGPT experiment, we train 20 different algorithms and test with 100 randomly sampled test matrices. All GPU based experiments were done in NVIDIA RTX A-6000 and CPU based experiments were done in AMD EPYC 7713 64-Core Processor.

For some of our experiments we compute matrix inverse in the action. As we primarily deal with symmetric square matrices, we used Cholesky decomposition + torch.linalg.LUsolve to compute the matrix inverse. The time needed to find each algorithm is reported in Section D.5.

### 5.1 MATRIX SQUARE ROOT FOR RANDOM MATRICES

We first compute matrix square root and its inverse on random matrices. Fig. 2a shows computing matrix square root inverse for empirical covariances of CIFAR-10 (Krizhevsky et al., 2009) images,

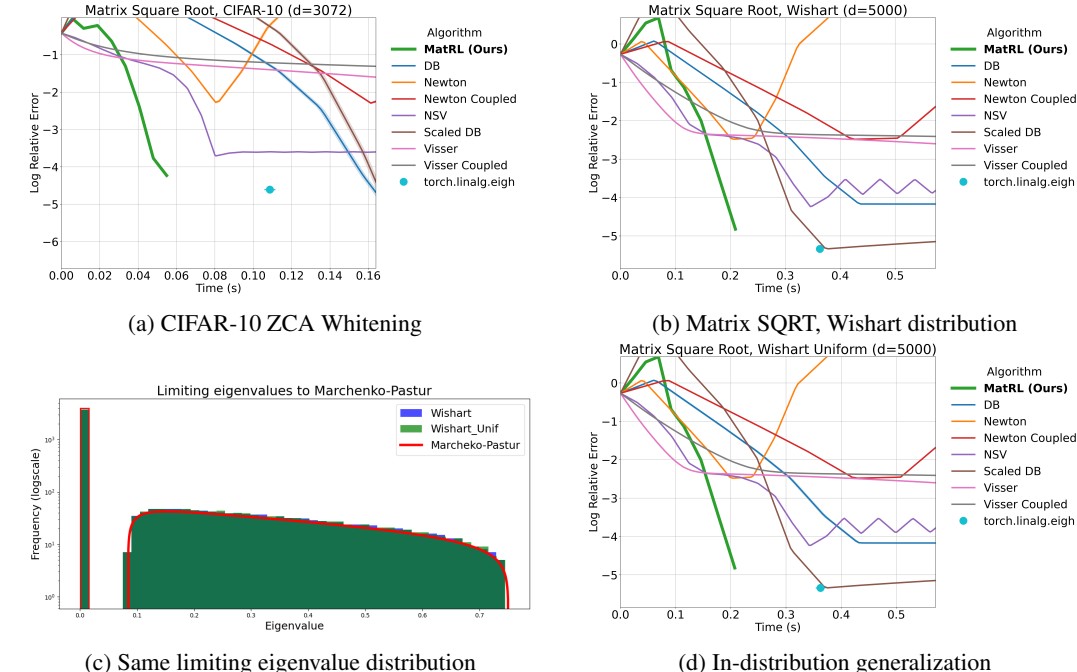

(a) CIFAR-10 ZCA Whitening

(b) Matrix SQRT, Wishart distribution

(c) Same limiting eigenvalue distribution

(d) In-distribution generalization

Figure 2: Experimental results of computing matrix square root for synthetic/real-world matrices. Each plot has existing iterative methods as baselines. Time for running torch.linalg.eigh is also included. At each timestep, we compute the sample mean and variance, and visualize the mean together with a shaded band corresponding to one standard deviation. We can see that the learned algorithm is faster than torch.linalg.eigh, but it sacrifices accuracy for speed. By testing the learned algorithm on the random matrix distribution with same limiting spectrum, we can verify Proposition 1.

which is used for ZCA whitening. Specifically, the random input matrix is $\hat{\Sigma} = \frac{1}{n}(X - \mu)^T(X - \mu)$ where $X \in \mathbb{R}^{n \times d}$ is a random batch of $n$ CIFAR-10 images, $\mu$ is the average of $n$ images, followed by normalizing by the Frobenius norm and adding $\epsilon I$ ($\epsilon = 1e - 3$). The second experiment in Fig. 2b computes matrix square root for Wishart matrix that is sampled from $A = \frac{X^\top X}{3d} + \epsilon I$ where $X \in \mathbb{R}^{d/4 \times d}$, $X_{ij} \sim \mathcal{N}(0, 1)$ i.i.d., $\epsilon = 1e - 3$. For both experiments there is substantial speedup compared to the baseline algorithms and torch.linalg.eigh. MatRL learned algorithm is $\approx$ 1.96x faster than torch.linalg.eigh for CIFAR-10 ZCA whitening, and $\approx$ 1.72x faster than the torch implementation of eigh for Wishart matrices, trading off accuracy for speed.

---

**Algorithm 1** Iterative SQRT for Wishart on GPU with $d = 5000$

---

**Input:** $A$
Initialize $X_0 = A, Y_0 = I$
Set $a \leftarrow [0.766, 1.876, 4.251, 0.924, 1], b \leftarrow [2.930, 0.819, 3.223, 0.460, 0.877]$,
// rounded off to three digits
**for** $i = 1$ to 5 **do**
    $X_i = a_{i-1}X_{i-1} + b_{i-1}(A - X_{i-1}^2)$
    $Y_i = a_{i-1}Y_{i-1} + b_{i-1}(I - Y_{i-1}X_{i-1})$
**end for**
**for** $i = 6$ to 8 **do**
    $X_i = 0.5(3I - X_{i-1}Y_{i-1})X_{i-1}$
    $Y_i = 0.5Y_{i-1}(3I - X_{i-1}Y_{i-1})$
**end for**
**return** $X_8$

---

Algorithm 1 shows the iterative square root algorithm for Wishart random matrices found by MatRL. Here we see a tendancy that during the early iterations, the algorithm prefers Visser iteration, whereas for latter steps the algorithm prefers coupled Newton-Schulz iteration. An intuition for such mixed iteration is as follows: when we see the loss curve, we can notice that the Visser iteration converges fast at first, but it quickly stabilizes and becomes very slow. Hence MatRL learns to use the cheap iterations at first to find a good initial start for NewtonSchulz, and run NewtonSchulz from that initial parameter for faster convergence.

At last, we verify the generalization guarantee that we had in Proposition 1. We test the algorithm in Algorithm 1 to a different matrix distribution with the same limiting spectrum, where each entries of $X$ are sampled i.i.d. from $\mathrm{Unif}[-\sqrt{3}, \sqrt{3}]$ instead of $\mathcal{N}(0,1)$. We denote the distribution as WishartUnif. In this case, the two spectrum converges to the same spectrum in distribution due to Marchenko-Pastur (Fig. 2c). Comparing the two curves in Fig. 2b and Fig. 2d shows that the learned algorithm generalizes well to different random matrix distributions with the same limiting spectrum.

## 5.2 COMPUTING FASTER MATRIX SIGN FOR NANOGPT MOMENTUM STATES

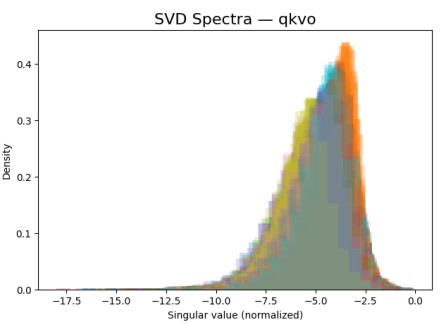

(a) Spectrum of QKVO momentum states

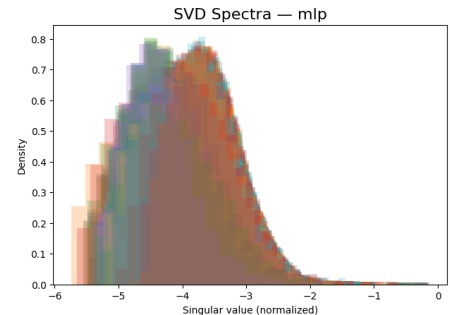

(b) Spectrum of MLP momentum states

Figure 3: Spectrum of momentum states for NanoGPT training. We can see that the spectrum of QKVO momentum states are similar, and the spectrum of MLP momentum states are similar.

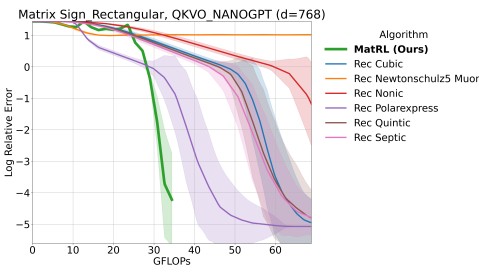

(a) Matrix Sign, QKVO momentum states

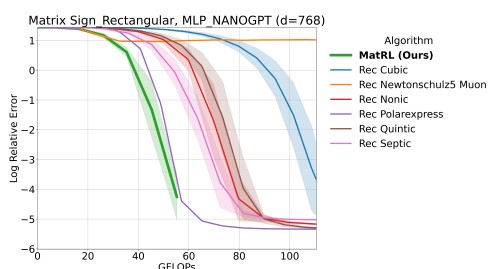

(b) Matrix Sign, MLP momentum states

Figure 4: Time versus loss curves for matrix sign. Each subplot compares iterative algorithms as baselines. At each timestep, we compute the sample mean and variance, and visualize the mean together with a shaded band corresponding to one standard deviation. For matrix sign, the discovered algorithm shows speedup, with variance due to diversity in the spectrum for optimizer states.

For NanoGPT momentum states, we run the NanoGPT training repository (Jordan et al. (2024a)) and store the momentum state every 125 steps, and split them randomly as train/test data. Surprisingly, the spectrum of QKVO layer momentum states and MLP layer momentum states are similar, respectively (see Fig. 3a, Fig. 3b), which implies a similar generalization phenomenon for the found algorithm. One practical issue was that the maximum eigenvalue deviated from matrix to matrix which led to poor generalization. To solve this issue, when we initialize $s_{root}$, we first initialized with the singular values of $A$ and then changed the 100 largest singular values with $np.linspace(0.1, 1, 100)$. Also, each action has different cost between MLP layer and QKVO layer

matrices due to their shape difference. Hence we separate the two and obtain different algorithms. The matrix size was too small to measure each action's runtime accurately, so we used GFLOPs as the measure of complexity instead.

We can see that the found algorithm is better than any odd polynomial baselines, and even better than PolarExpress(Amsel et al., 2025), an optimally computed quintic polynomial iteration to approximate matrix sign. The major gain of MatRL here is finding better algorithms by expanding the action space: we do not stick to quintic polynomials only, but look at other iteration types such as cubic, quintic, septic, and nonic iterations. It turns out that for QKVO layers, using well-tuned cubic polynomials can be better than quintic polynomials in terms of FLOPs. For MLP layers using higher-order polynomials can be beneficial because for $\mathbb{R}^{m \times n}$ for $m < n$, the number of FLOPs to compute a $2k + 1$ - degree polynomial becomes $4nm^2 + 2(k-1)m^3$ and the efficiency of each iteration depends on the ratio of $m$ and $n$. The found algorithm uses nonic, septic, and cubic polynomial iterations, see Section D.3 for details. Finally, Section D.4 shows that the learned algorithm generalizes to different sizes of NanoGPT training.

### 5.3 Different computation environments yield different algorithms

The mixing tendency appears for different matrix functions as well. Preference for a certain iteration over another emerges from two axes, the time it takes for an iteration and how effectively the iteration decreases the loss. Recalling the motivating example in Table 1, different computing environments, e.g. hardware (CPU/GPU), precision (Single/Double), or even the size of the matrix can decide the best algorithm. Here we only demonstrate how precision can change optimal algorithms. A full list of different setups and found algorithms are in Section D.

---

**Algorithm 2** SIGN on GPU (FLOAT)

1: **Input:** $A$
2: Initialize $X_0 = A$
3: $a \leftarrow [35.536]$
4: $b \leftarrow [0.094, 1.607, 1.439, 1.191, 1.029, 1]$
5: **for** $i = 1$ to $1$ **do**
6: $\quad X_i = 0.5(a_{i-1}X_{i-1} + (a_{i-1}X_{i-1})^{-1})$
7: **end for**
8: **for** $i = 2$ to $7$ **do**
9: $\quad X_i = 1.5(b_{i-2}X_{i-1}) - 0.5(b_{i-2}X_{i-1})^3$
10: **end for**
11: **return** $X_7$

---

**Algorithm 3** SIGN on GPU (DOUBLE)

1: **Input:** $A$
2: Initialize $X_0 = A$
3: $a \leftarrow [22.055, 0.244, 0.590, 0.938, 0.998, 1]$

4: **for** $i = 1$ to $6$ **do**
5: $\quad X_i = 0.5(a_{i-1}X_{i-1} + (a_{i-1}X_{i-1})^{-1})$
6: **end for**
7: **return** $X_6$

---

When the precision is double, the relative runtime ratio between inverse and matrix multiplication is not as high as that of single precision. Hence, the model prefers Newton's method more for double precision case, and Algorithm 3 does not use NewtonSchulz iterations.

## 6 Conclusion

In this paper we propose MatRL, an MCTS-based automated solution to find iterative algorithms for matrix function computation. We showed that we can generate an algorithm specifically tailored for a specific input matrix distribution and compute environment, and the found algorithm is guaranteed to generalize to different matrix distributions with the same limiting spectrum under certain assumptions. We verify our findings with experiments, showing MatRL found algorithms that are faster than existing baselines and competitive to standard torch library.

Our work has many future directions. One direction is overcoming the current limitations of our work, e.g. extending the result to rectangular and sparse matrices. Another interesting direction is not fixing the matrix iterations beforehand, but making the RL agent discover novel iterations that can ensure stability, such as the Denman-Beavers iteration. At last, automatic discovery of optimal parameters of optimization algorithms such as Muon (Jordan et al., 2024b) with a different reward (e.g. validation error) could be an interesting direction.

ETHICAL STATEMENT

The authors have no ethical concerns regarding the paper.

REPRODUCIBILITY STATEMENT

Also, the experimental results given in Section 5 and Section C are reproducible by running the code in the supplementary material.

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

## A  TECHNICAL PROOFS

We first show that the environment can be parametrized by the spectrum.

**Proposition A.2.** *Let $\{f_k(X, Y, A)\}_{k=1}^m$ is a set of congruence invariant diagonal preserving matrix functions that take $X, Y, A$ as input and outputs $(X', Y')$, $\mathcal{L}(X, A)$ be a congruence invariant matrix function that takes $X, A$ as input and outputs a scalar, and $X_0, Y_0, \{X_k\}_{k=1}^{m+1}$ satisfy*

$$(X_0, Y_0) = (A, I) \quad or \quad (I, A)$$

*and*

$$(X_{k+1}, Y_{k+1}) = f_k(X_k, Y_k, A), \quad k \in [m].$$

*At last, write $A = U D_A U^T$. Then, the following properties hold:*
*i) $X_k, Y_k \sim A$ for all $k \in [m+1]$.*
*ii) Write $X_k = U P_k U^T$, $Y_k = U Q_k U^T$. Then, $P_{k+1}$, $Q_{k+1}$ depends only on $P_k$, $Q_k$ and $D_A$.*
*iii) The loss $\mathcal{L}(X_k, A)$ only depends on $P_k, D_A$.*

*Proof.* i) We prove by induction.
If $k = 0$, we know that $X_0, Y_0 = A, I$ or $I, A$ hence they are similar with $A$.
Say $X_t, Y_t$ are similar with $A$. Then $X_t = U D_1 U^T, Y_t = U D_2 U^T$ for diagonal $D_1, D_2$. Now we can see that

$$f_t(X_t, Y_t, A) = f_t(U D_1 U^T, U D_2 U^T, U D_A U^T) = U f_t(D_1, D_2, D_A) U^T = (U D_1' U^T, U D_2' U^T),$$

for some diagonal $D_1', D_2'$. The second equality comes from congruent invariance, and the third equality follows from diagonal preservence. Hence, $X_{t+1} = U D_1' U^T$ and $X_{t+1}$ and $A$ are similar, $Y_{t+1} = U D_2' U^T$ and $Y_{t+1}$ and $A$ are also similar.
ii) From the proof of i) we know that $f_t(D_1, D_2, D_A) = (D_1', D_2')$, where $D_1, D_2, D_A, D_1', D_2'$ are the spectrum of $X_t, Y_t, A, X_{t+1}, Y_{t+1}$, respectively.
iii) We know $X_k$ and $A$ are similar. Let $X_k = U P_k U^T$ and $A = U D_A U^T$. $\mathcal{L}(U P_k U^T, U D_A U^T) = \mathcal{L}(P_k, D_A)$ by congruent invariance. $\qquad\square$

Next we show that all iterations that we deal with in the paper is congruent invariant diagonal preserving. See Section B.6 and Table 6 for the iterations of interest.

**Lemma A.1.** *Assume $f(X_1, X_2, \cdots X_k)$, $g(X_1, X_2, \cdots X_k)$ are congruent invariant diagonal preserving. Then, $f + g$, $fg$, $f^{-1}$ are all congruent invariant diagonal preserving.*

*Proof.* Diagonal preserving is simple, as sum, multiple, inverse of diagonal matrices are diagonal. Let's show congruence invariance.

$$f(Q X_1 Q^T, \cdots Q X_k Q^T) + g(Q X_1 Q^T, \cdots Q X_k Q^T) = Q f(X_1, \cdots X_k) Q^T + Q g(X_1, \cdots X_k) Q^T$$

and

$$Q f(X_1, \cdots X_k) Q^T + Q g(X_1, \cdots X_k) Q^T = Q(f(X_1, \cdots X_k) + g(X_1, \cdots X_k)) Q^T.$$

$$f(Q X_1 Q^T, \cdots Q X_k Q^T) g(Q X_1 Q^T, \cdots Q X_k Q^T) = Q f(X_1, \cdots X_k) Q^T Q g(X_1, \cdots X_k) Q^T$$

and

$$Q f(X_1, \cdots X_k) Q^T Q g(X_1, \cdots X_k) Q^T = Q(f(X_1, \cdots X_k) g(X_1, \cdots X_k)) Q^T.$$

$$f(Q X_1 Q^T, \cdots, Q X_k Q^T)^{-1} = (Q f(X_1, \cdots X_k) Q^T)^{-1} = Q f(X_1, \cdots X_k)^{-1} Q^T.$$

$\qquad\square$

**Proposition A.3.** *Say $f(X_1, X_2, \cdots X_k)$ is a rational function of $X_1, \cdots X_k$, i.e. $f(X_1, X_2, \cdots X_k) = P(X_1, \cdots X_k) Q(X_1, \cdots X_k)^{-1}$ for polynomials $P, Q$. Then $f$ is congruent invariant diagonal preserving.*

*Proof.* We know that $P, Q$, and hence $PQ^{-1}$ is congruent invariant diagonal preserving from lemma 1. $\qquad\square$

As all iterations in Table 6 or Section B.6 are rational functions, the iterations of interest are congruent invariant diagonal preserving.

At last we show Proposition 1: our found algorithm will generalize to the matrices drawn from the distribution with the same limiting eigenvalue spectrum.

**Proposition A.4.** *(Generalization of the discovered algorithm, Proposition 1 of the main paper) Say we have a sequence of symmetric random matrix distributions $\mathcal{P}_m, \mathcal{Q}_m$ defined in $\mathbb{R}^{m \times m}$, and denote random matrices sampled from $\mathcal{P}_m, \mathcal{Q}_m$ as $X, Y$. Let the empirical eigenvalue value distribution of $X \sim \mathcal{P}_m, Y \sim \mathcal{Q}_m$ be $\mu_m(X), \nu_m(Y)$, and their support be $S_m, S'_m$, respectively. Now, suppose*
*(i) (Identical limiting distribution)*

$$\mathbb{P}(\mu_m(X) \Rightarrow \mu^*) = \mathbb{P}(\nu_m(Y) \Rightarrow \mu^*) = 1,$$

*i.e. both $\mu_m(X)$ and $\nu_m(Y)$ converges weakly to a common distribution $\mu^*$ with probability 1.*
*(ii) (Interval support of the limiting distribution) The support of $\mu^*$ is an interval $[a, b]$.*
*(iii) (Convergence of support) We have*

$$\lim_{m \to \infty} \mathbb{P}(S_m \subseteq [a - \epsilon, b + \epsilon]) = \lim_{m \to \infty} \mathbb{P}(S'_m \subseteq [a - \epsilon, b + \epsilon]) = 1,$$

*for all $\epsilon > 0$.*
*With the assumptions above, let $f$ be the matrix function we would like to compute, $f_k^*$ the step $k$ transformation of eigenvalues of the algorithm found by MatRL, and $\mathcal{L}$ be the loss. Assume $f, f_k^*, L$ are continuous in $[a - \epsilon_0, b + \epsilon_0]$ for some $\epsilon_0 > 0$. Write the empirical loss of the random matrix $X$ as*

$$\mathcal{L}_k(X) = \frac{1}{m} \sum_{i=1}^{m} L(f(\lambda_i), f_k^*(\lambda_i)),$$

*where $\lambda_i$ are eigenvalues of $X$.*
*Then, there exists $M_{\epsilon, \delta}$ such that*

$$m \geq M_{\epsilon, \delta} \Rightarrow \mathbb{P}_{X \sim \mathcal{P}_m, Y \sim \mathcal{Q}_m}[|\mathcal{L}_k(X) - \mathcal{L}_k(Y)| < \epsilon] \geq 1 - \delta.$$

*Proof.* We write

$$\mathcal{L}^* = \int L(f(\sigma), f_k(\sigma)) d\mu^*(\sigma).$$

We would like to show that for sufficiently large $m$, $|\mathcal{L}_k(X) - \mathcal{L}^*| < \epsilon/2$ with high probability. As $\mathbb{P}(\mu_m(X) \Rightarrow \mu^*) = 1$, we know that with probability 1,

$$\lim_{m \to \infty} \int f d\mu_m(X) = \int f d\mu^*$$

for all continuous bounded $f$. We shall extend $L(f(x), f_k(x))$ in a way that it is continuous bounded in $\mathbb{R}$ and the "difference" is small.
First, we know that $L(f(x), f_k(x))$ is continuous in $[a - \epsilon_0, b + \epsilon_0]$. Say

$$A = \max_{x \in [a - \epsilon_0, b + \epsilon_0], y \in [a, b]} |L((f(x), f_k(x))| + |L((f(y), f_k(y))|.$$

Now, choose $\epsilon' = \min\{\epsilon_0, \epsilon/8A\}$ (when $A = 0$ we just have $\epsilon' = \epsilon_0$). With the chosen $\epsilon'$, define $\tilde{L}$ as

$$\tilde{L}(x) = \begin{cases} L(f(x), f_k(x)) & if \quad x \in [a, b] \\ (x - a + \epsilon') \frac{L(f(a), f_k(a))}{\epsilon'} & if \quad x \in [a - \epsilon', a] \\ (-x + b + \epsilon') \frac{L(f(b), f_k(b))}{\epsilon'} & if \quad x \in [b, b + \epsilon'] \\ 0 & if \quad x \in (-\infty, a - \epsilon'], [b + \epsilon', \infty), \end{cases}$$

which is a bounded continuous function in $\mathbb{R}$. At last, choose $M_1$ sufficiently large so that $m \geq M_1$ implies

$$|\int \tilde{L} d\mu_m(X) - \int \tilde{L} d\mu^*| < \epsilon/4$$

with probability at least $1 - \delta/4$ and

$$\mathbb{P}(S_m \subseteq [a - \epsilon', b + \epsilon']) \geq 1 - \delta/4.$$

Such $m$ exists because of assumptions (i) and (iii). Now, we know that

$$\mathcal{L}_k(X) = \int L(f(x), f_k(x))\mu_m(X).$$

Moreover, the function $|\tilde{L}(x) - L(f(x), f_k(x))| \leq A$ for $x \in S_m$. This is because

$$|\tilde{L}(x)| \leq \max_{x \in [a,b]} |L(f(x), f_k(x))|, \quad S_m \subseteq [a - \epsilon', b + \epsilon'] \subseteq [a - \epsilon_0, b + \epsilon_0].$$

As $\tilde{L}(x) - L(f(x), f_k(x)) = 0$ for $x \in [a, b]$, the value

$$|\int \tilde{L} d\mu_m(X) - \int L(f(x), f_k(x)) d\mu_m(X)| \leq 2\epsilon' A \leq \epsilon/4$$

with probability at least $1 - \delta/2$. Hence, when $m \geq M_1$,

$$|\mathcal{L}_k(X) - \int \tilde{L} d\mu^*| = |\mathcal{L}_k(X) - \mathcal{L}^*| < \epsilon/2$$

with probability at least $1 - \delta/2$. We can do the same argument for $Y$ to find $M_2$. Take $M_{\epsilon,\delta} = \max\{M_1, M_2\}$. Using union bound, we can see the probability that both $|\mathcal{L}_k(X) - \mathcal{L}^*| < \epsilon/2$ and $|\mathcal{L}_k(Y) - \mathcal{L}^*| < \epsilon/2$ happens is at least $1 - \delta$. Hence, $\mathbb{P}_{X \sim \mathcal{P}_m, Y \sim \mathcal{Q}_m}[|\mathcal{L}_k(X) - \mathcal{L}_k(Y)| < \epsilon] \geq 1 - \delta$. $\qquad \square$

## B    LIST OF USED MATRIX ITERATIONS

We first present a table that shows different types of baseline algorithms used in the paper with references.

Table 2: List of baselines

| Matrix function | List of baselines |
| --- | --- |
| Inverse | NS (Pan & Schreiber, 1991), Chebyshev (Li et al., 2011) |
| Sign | Newton (Higham, 2008), NS (Schulz, 1933), ScaledNewton (Byers & Xu, 2008), ScaledNS (Chen & Chow, 2014), Halley(Nakatsukasa et al., 2010) |
| Square root | DB(Denman & Beavers Jr, 1976), NSV(Higham, 1997)(2.6), Scaled DB(Higham, 1997), Visser(Higham, 2008), Newton(Hoskins & Walton, 1979) |
| 1/3 - root | Iannazzo (Iannazzo, 2006), Visser (Higham, 2008), Newton (Iannazzo, 2006)(1.2) |
| Rectangular Sign | NS (Schulz, 1933), NewtonSchulz5 Muon (Jordan et al., 2024b), PolarExpress (Amsel et al., 2025), Higher order approximations of the sign function |

### B.1    ITERATIVE METHODS ASSOCIATED WITH INVERSE

We have two different baselines for inverse. One is Newton's method proposed by Schulz, which is the iteration

$$\textbf{(InvNewton)} \qquad X_{k+1} = 2X_k - X_k A X_k.$$

For an appropriate initialization, the norm $\|I - AX_k\|_2$ will converge quadratically to zero. This is because we can write

$$I - AX_{k+1} = I - 2AX_k + AX_k AX_k = (I - AX_k)^2.$$

Another baseline is applying Chebyshev's iteration to the function $X^{-1} - A$. We have

$$\textbf{(InvChebyshev)} \qquad X_{k+1} = 3X_k - 3X_k A X_k + X_k A X_k A X_k.$$

With similar logic we can obtain $I - AX_{k+1} = (I - AX_k)^3$. Hence at each iteration the error decreases cubically. The drawback is that Chebyshev's method needs at least three matrix-matrix multipications each iteration.

### B.2    ITERATIVE METHODS ASSOCIATED WITH SIGN

The simplest method to compute matrix sign is Newton's method, where the iteration is given as

$$\textbf{(SignNewton)} \qquad X_{k+1} = \frac{1}{2}(X_k + X_k^{-1}).$$

The NewtonSchulz variant avoids computing inverse by using the iteration

$$X_{k+1} = \frac{1}{2}(3X_k - X_k X_k^T X_k),$$

hence for symmetric matrices

$$\textbf{(SignNewtonSchulz)} \qquad X_{k+1} = \frac{1}{2}(3X_k - X_k^3).$$

Newton's method has scaled variants, where we do

$$\textbf{(SignScaledNewton)} \qquad X_{k+1} = \frac{1}{2}(\mu_k X_k + (\mu_k X_k)^{-1}),$$

for specific $\mu_k$. Our baseline is the one proposed in (Byers & Xu, 2008). Here $\mu_k$ is defined as the following: we let $a, b$ be constants that satisfy $a \leq \sigma_n \leq \sigma_1 \leq b$ for the singular values of $A$. Then

$$\mu_0 = \frac{1}{\sqrt{ab}}, \quad \mu_1 = \sqrt{\frac{2}{\sqrt{a/b} + \sqrt{b/a}}}, \quad \mu_k = \sqrt{\frac{2}{\mu_{k-1} + \mu_{k-1}^{-1}}}, k \geq 2.$$

$a$ and $b$ can be obtained by computing $\|A\|_2$ and $\|A^{-1}\|_2^{-1}$. NewtonSchulz method may also have variants: a recent variant in (Chen & Chow, 2014) scales each $X_k$ as

$$\textbf{(SignScaledNewtonSchulz)} \qquad X_{k+1} = \frac{3}{2}\rho_k X_k - \frac{1}{2}(\rho_k X_k)^3,$$

where $X_0 = A/\lambda_{|max|}(A)$, $x_0 = \lambda_{|min|}(A)/\lambda_{|max|}(A)$ and

$$\rho_k = \sqrt{\frac{3}{1 + x_0 + x_0^2}}, \quad x_{k+1} = \frac{1}{2}\rho_k x_k(3 - \rho_k^2 x_k^2).$$

Halley's method uses a rational approximation of sign function to compute the matrix sign. The iteration is written as

$$\textbf{(SignHalley)} \qquad X_{k+1} = X_k(a_k I + b_k X_k^2)(I + c_k X_k^2)^{-1},$$

where default Halley's iteration uses $a = c = 3, b = 1$ and the scaled Halley in (Nakatsukasa et al., 2010) uses certain optimal coefficients.

Newton variant is essentially a variant of Newton's method where we do

$$\textbf{(SignNewtonVariant)} \qquad X_{k+1} = 2X_k(I + X_k^2)^{-1}.$$

This is the inverse of SignNewton, and $X_k$ converges to $\text{sign}(A)^{-1}$, which is $\text{sign}(A)$ when $A$ is invertible.

### B.3 ITERATIVE METHODS ASSOCIATED WITH SQUARE ROOT

The simplest method in this case is also the Newton's method,

$$\textbf{(SqrtNewton)} \qquad X_{k+1} = \frac{1}{2}(X_k + X_k^{-1}A).$$

The above method can be unstable, which led to the development of coupled iterations. Denman-Beavers iteration uses the following coupled iterationn of $X_k$ and $Y_k$: Denman-Beavers is initialized with $X_0 = A, Y_0 = I$ and iteratively applies

$$\textbf{(SqrtDenmanBeavers)} \qquad \begin{cases} X_{k+1} = \frac{1}{2}(X_k + Y_k^{-1}) \\ Y_{k+1} = \frac{1}{2}(Y_k + X_k^{-1}). \end{cases}$$

Here, $X_k \to A^{1/2}$ and $Y_k \to A^{-1/2}$. There is a variant of Denman-Beavers that avoids computing matrix inverse - introduced in (Higham, 1997), the iteration writes

$$\textbf{(SqrtNewtonSchulzVariant)} \qquad \begin{cases} X_{k+1} = \frac{1}{2}(3X_k - X_k Y_k X_k) \\ Y_{k+1} = \frac{1}{2}(3Y_k - Y_k X_k Y_k). \end{cases}$$

Like ScaledNewton, we have a scaled variant of Denman-Beavers. The scaling we use is a variant of Byer's scaling (Byers, 1987) introduced in (Higham, 1997). The iteration is given as

$$\textbf{(SqrtScaledDenmanBeavers)} \qquad \begin{cases} \gamma_k = |\det X_k \det Y_k|^{-1/2n} \\ X_{k+1} = \frac{1}{2}(\gamma_k X_k + (\gamma_k Y_k)^{-1}) \\ Y_{k+1} = \frac{1}{2}(\gamma_k Y_k + (\gamma_k X_k)^{-1}). \end{cases}$$

The cost of computing $\gamma_k$ is negligible when we use decomposition methods such as LU decomposition or Cholesky to compute matrix inverse.

At last, there is the fixed-point iteration, which we will denote as the Visser iteration (Higham, 2008). The Visser iteration is given as

$$\textbf{(SqrtVisser)} \qquad X_{k+1} = X_k + \frac{1}{2}(A - X_k^2).$$

### B.4 ITERATIVE METHODS ASSOCIATED WITH 1/3-ROOT

There are a number of stable methods to compute matrix $p$-th root (see (Iannazzo, 2006) for different methods). We use the following method as a baseline: initialize $X_0 = I, Y_0 = A$ and

$$\textbf{(prootIannazzo)} \qquad \begin{cases} X_{k+1} = X_k \left(\frac{2I+Y_k}{3}\right) \\ Y_{k+1} = \left(\frac{2I+Y_k}{3}\right)^{-3} Y_k \end{cases}$$

With this iteration, $X_k \to A^{1/3}$ and $Y_k \to I$. We have Newton's method and Visser's iteration as we had for square root:

$$\textbf{(prootNewton)} \qquad X_{k+1} = (2X_k + X_k A^{-2})/3$$

is the Newton's method, and

$$\textbf{(prootVisser)} \qquad X_{k+1} = X_k + \frac{1}{3}(A - X_k^3)$$

becomes Visser's iteration.

### B.5 ITERATIVE METHODS ASSOCIATED WITH RECTANGULAR SIGN

Computing matrix sign for rectangular matrices in a fast reliable way became important due to the recent Muon optimizer. We can use methods such as signscalednewtonschulz, which can be written as

$$\textbf{(NewtonSchulz)} \quad X_{k+1} = \frac{3}{2}X_k - \frac{1}{2}X_k X_k^T X_k$$

Some higher order approximations for matrix sign function are

$$\textbf{(Quintic)} \quad X_{k+1} = 1.875X_k - 1.25X_k X_k^T X_k + 0.375(X_k X_k^T)^2 X_k$$

$$\textbf{(Septic)} \quad X_{k+1} = 2.405X_k - 3.0605X_k X_k^T X_k + 2.405(X_k X_k^T)^2 X_k - 0.75(X_k X_k^T)^3 X_k$$

$$\textbf{(Nonic)} \quad X_{k+1} = 2.25X_k - 2.4375X_k X_k^T X_k + 1.6875(X_k X_k^T)^2 X_k - 0.5625(X_k X_k^T)^3 X_k + 0.0625(X_k X_k^T)^4 X_k$$

There has been some variants of quintic newton-schulz iteration. In the newtonschulz module in the Muon optimizer they use

$$\textbf{(Newtonschulz5Muon)} \quad X_{k+1} = 3.4445X_k - 4.7750X_k X_k^T X_k + 2.0315(X_k X_k^T)^2 X_k$$

In PolarExpressAmsel et al. (2025), a series of optimal quintic coefficients are given.

$$\textbf{(PolarExpress)} \quad X_{k+1} = a_k X_k - b_k X_k X_k^T X_k + c_k (X_k X_k^T)^2 X_k$$

for precomputed $a_k, b_k, c_k$.

## B.6 SUMMARY

We present a table of the matrix functions and iterations that we used.

Table 3: Iterative methods for computing matrix inverse, sign, square root, and 1/3-root.

| Method | Iteration Formula |
|---|---|
| **Methods for Inverse** | |
| Newton (Schulz) | $X_{k+1} = 2X_k - X_k A X_k$ |
| Chebyshev | $X_{k+1} = 3X_k - 3X_k A X_k + X_k A X_k A X_k$ |
| **Methods for Sign** | |
| Newton | $X_{k+1} = \frac{1}{2}(X_k + X_k^{-1})$ |
| NewtonSchulz | $X_{k+1} = \frac{1}{2}(3X_k - X_k^3)$ |
| ScaledNewton | $X_{k+1} = \frac{1}{2}(\mu_k X_k + (\mu_k X_k)^{-1})$ |
| ScaledNewtonSchulz | $X_{k+1} = \frac{3}{2}\rho_k X_k - \frac{1}{2}(\rho_k X_k)^3$ |
| Halley | $X_{k+1} = X_k(a_k I + b_k X_k^2)(I + c_k X_k^2)^{-1}$ |
| NewtonVariant | $X_{k+1} = 2X_k(I + X_k^2)^{-1}$ |
| **Methods for SquareRoot** | |
| Newton | $X_{k+1} = \frac{1}{2}(X_k + X_k^{-1}A)$ |
| DenmanBeavers | $\begin{cases} X_{k+1} = \frac{1}{2}(X_k + Y_k^{-1}) \\ Y_{k+1} = \frac{1}{2}(Y_k + X_k^{-1}) \end{cases}$ |
| NewtonSchulzVariant | $\begin{cases} X_{k+1} = \frac{1}{2}(3X_k - X_k Y_k X_k) \\ Y_{k+1} = \frac{1}{2}(3Y_k - Y_k X_k Y_k) \end{cases}$ |
| ScaledDenmanBeavers | $\begin{cases} \gamma_k = \lvert \det X_k \det Y_k \rvert^{-1/2n} \\ X_{k+1} = \frac{1}{2}(\gamma_k X_k + (\gamma_k Y_k)^{-1}) \\ Y_{k+1} = \frac{1}{2}(\gamma_k Y_k + (\gamma_k X_k)^{-1}) \end{cases}$ |
| Visser | $X_{k+1} = X_k + \frac{1}{2}(A - X_k^2)$ |
| **Methods for 1/3-th Root** | |
| Iannazzo | $\begin{cases} X_{k+1} = X_k\left(\frac{2I + Y_k}{3}\right) \\ Y_{k+1} = \left(\frac{2I + Y_k}{3}\right)^{-3} Y_k \end{cases}$ |
| Newton | $X_{k+1} = \frac{1}{3}(2X_k + X_k A^{-2})$ |
| Visser | $X_{k+1} = X_k + \frac{1}{3}(A - X_k^3)$ |
| **Methods for Rectangular Sign** | |
| Cubic | $X_{k+1} = \frac{3}{2}X_k - \frac{1}{2}X_k X_k^T X_k$ |
| Quintic | $X_{k+1} = 1.875X_k - 1.25X_k X_k^T X_k + 0.375(X_k X_k^T)^2 X_k$ |
| Septic | $X_{k+1} = 2.405X_k - 3.0605X_k X_k^T X_k + 2.405(X_k X_k^T)^2 X_k - 0.75(X_k X_k^T)^3 X_k$ |
| Nonic | $X_{k+1} = 2.25X_k - 2.4375X_k X_k^T X_k + 1.6875(X_k X_k^T)^2 X_k - 0.5625(X_k X_k^T)^3 X_k + 0.0625(X_k X_k^T)^4 X_k$ |
| NewtonSchulz5Muon | $3.4445X_k - 4.7750X_k X_k^T X_k + 2.0315(X_k X_k^T)^2 X_k$ |
| PolarExpress | $a_k X_k - b_k X_k X_k^T X_k + c_k(X_k X_k^T)^2 X_k$ |

## C  EXPERIMENTAL DETAILS

### C.1  BASICS OF MONTE-CARLO TREE SEARCH

Suppose we have a deterministic environment $\mathcal{E}$ which is defined by a 5-tuple, $(\mathcal{S}, \mathcal{A}, \mathcal{T}, r, t)$. $\mathcal{S}$ denotes the set of states, $\mathcal{A}$ denote the set of actions that one can take in each state, $\mathcal{T} : \mathcal{S} \times \mathcal{A} \to \mathcal{S}$ gives how state transition occurs from state $s$ when we apply action $a$, $r : \mathcal{S} \times \mathcal{A} \to \mathbb{R}$ gives the amount of reward one gets when we do action $a$ at state $s$, and $t : \mathcal{S} \to \{0, 1\}$ denotes whether the state is terminal or not. Monte-carlo tree search enables us to find the optimal policy (Browne et al., 2012) $\pi : \mathcal{S} \to \mathcal{A}$ that decides which action to take at each state to maximize the reward over the trajectory that $\pi$ generates.

The basic idea is to traverse over the "search tree" in an asymmetric manner, using the current value estimation of each state. Each node of the search tree has a corresponding state $s$, its value estimation $V_s$, and visit count $N_s$. The algorithm is consisted of four steps, the selection step, the expansion step, the simulation step, and the backpropagation step.

**During the selection step,** the algorithm starts at the root node $s_0$ and selects the next node to visit using $V_{\mathcal{T}(s,a)}$ and $N_{\mathcal{T}(s,a)}$. One widely-used method is using the upper confidence bounds for trees (UCT) (Kocsis & Szepesvári, 2006). At node $s$, the UCT is defined by

$$V_{\mathcal{T}(s,a)} + C\sqrt{\frac{\log(N_s)}{N_{\mathcal{T}(s,a)}}},$$

where $C$ determines the exploration-exploitation tradeoff. The algorithm selects the next state $\mathcal{T}(s, a)$ that maximizes UCT, and such selection continues until the algorithm meets a node of which not all child nodes have been visited.

**In the expansion step,** the algorithm adds a child node that has not been visited to the search tree.

**In the simulation step,** the algorithm starts from the added node in the expansion step and uses a default policy to generate a trajectory until they meet a termination criterion. The default policy can either be a random policy or handcrafted heuristics (Chaslot et al., 2008).

**Finally, in the backpropagation step,** each value estimation $V_s$ on the trajectory is updated.

Monte-Carlo tree search gained its popularity to obtain strategies for games such as Tictactoe (Veness et al., 2011) or Go (Silver et al., 2017), (Hoock et al., 2010), as well as real-time stategic games (Soemers, 2014). Not only that, the algorithm was also applied to combinatorial optimization problems (Sabharwal et al., 2012), (Rimmel et al., 2011), symbolic regression (Kamienny et al., 2023), and complex scheduling problems (Chaslot et al., 2006), (Matsumoto et al., 2010), (Li et al., 2021) - which is most relevant to our work.

### C.2  DETAILED EXPLANATION OF MATRL

MatRL is summarized in Section 3.3. Here we explain how each subroutine $Expandable$, $Best_{UCB}$, $ExpandNode$, $SampleRolloutList$, and $backpropagate$ is implemented.

To begin with, we have two important flags at each state. One flag is IsTransitionable: if the iteration type and all parameters for that iteration is fixed, we set IsTransitionable(s) = True. Else, IsTransitionable(s) = False. Another flag is IsCoupled: for the root state, IsCoupled = True. If you use a coupled iteration at a state where IsCoupled = True, IsCoupled = True at the next state also. If you use an iteration that is not coupled, we set IsCoupled = False. When IsCoupled = True, you can do either coupled or uncoupled iteration. When IsCoupled = False, you can either do the "coupling" iteration (that would be specified later for each matrix function) or an iteration that is not coupled. Whenn you do the coupling iteration, IsCoupled = True for the next state, else IsCoupled = False. A table that summmarizes the transition of IsCoupled is as below.

$Expandable(s)$ is a method that determines whether it is possible to expand a child node from current node $s$. If $IsTransitionable(s) = True$, the possible choice of next action becomes a discrete set of iterations. Hence $Expandable(s) = True$ if $IsCoupled(s) = False$ and number of children of $s$ < number of iterations that are not coupled, or $IsCoupled(s) = True$ andd number of children of $s$ < number of iterations - 1. We subtract 1 because we will not expand with coupling

Table 4: State transition of IsCoupled

| Current IsCoupled | Iteration Type | Next IsCoupled |
|---|---|---|
| True | Coupled iteration | True |
| True | Uncoupled iteration | False |
| False | Coupling iteration | True |
| False | Non-coupling iteration | False |

iteration. If $IsTransitionable = False$, we expand with a continuous variable hence we do progressive widening. If number of children of $s < C_{pw}N(s)^{\alpha_{pw}}$ where $n(s)$ is the number of visits for node $s$, we return True and else return False.

---

**Algorithm 4** $Expandable(s)$

---

1: **if** $IsTransitionable(s)$ **then**
2:    **if** $IsCoupled(s) =$ False **then**
3:       **if** number of children of $s <$ number of *non-coupled* iterations **then**
4:          **return** True
5:       **else**
6:          **return** False
7:       **end if**
8:    **else**
9:       **if** num_children$(s) <$ num_iterations $- 1$ **then**
10:         **return** True
11:       **else**
12:         **return** False
13:       **end if**
14:    **end if**
15: **else**
16:    **if** number of children of $s < C_{pw}n(s)^{\alpha_{pw}}$ **then**
17:       **return** True
18:    **else**
19:       **return** False
20:    **end if**
21: **end if**

---

$Best_{UCB}(s)$ is simple: Choose the child node $c$ with the maximal value of $V(c) + C_{ucb}\sqrt{\frac{\log n(s)+1}{n(c)}}$.

---

**Algorithm 5** $Best_{UCB}(s)$

---

1: **Input:** Node $s$ with children set $\mathcal{C}(s)$
2: **Parameters:** Exploration constant $C_{ucb}$
3: $best\_value \leftarrow -\infty$
4: $best\_child \leftarrow$ null
5: **for all** $c \in \mathcal{C}(s)$ **do**
6:    $score \leftarrow V(c) + C_{ucb}\sqrt{\frac{\log(n(s))+1}{n(c)}}$
7:    **if** $score > best\_value$ **then**
8:       $best\_value \leftarrow score$
9:       $best\_child \leftarrow c$
10:    **end if**
11: **end for**
12: **return** $best\_child$

---

$ExpandNode(s)$ depends on $IsTransitionable$. If $IsTransitionable(s) = True$, simply adding a node that hasn't been visited is enough, because the children are discrete. If $IsTransitionable(s) = False$, the children can take continuous parameters. If $num\_child(s) \leq$

$E$ for hyperparameter $E$, do random sampling in range $[lo, hi]$ that is prespecified. Else, find the child with the best value and sample near that parameter $p$. Specifically, with probability 0.05, sample uniformly at random from $[lo, hi]$. Else sample random uniform at a new interval $[lo, hi] \cap [p - stddev\_scale * (hi - lo)/2.0, p + stddev\_scale * (hi - lo)/2.0]$. $stddev\_scale = 1/\log(2 + n(s))$ decays logarithmically with $n(s)$, the visit count.

---

**Algorithm 6** $ExpandNode(s)$

---

1: **if** $IsTransitionable(s)$ **then**
2:     Add a new discrete child node to $s$
3: **else**
4:     **if** $num\_child(s) \leq E$ **then**
5:         Sample $x \sim \mathcal{U}[lo, hi]$
6:         Add child node with parameter $x$
7:     **else**
8:         $p \leftarrow$ parameter of best-value child of $s$
9:         $w \leftarrow (hi - lo)/2$
10:        $stddev\_scale \leftarrow 1/\log(2 + n(s))$
11:        $r \sim \mathcal{U}[0, 1]$
12:        **if** $r < 0.05$ **then**
13:           Sample $x \sim \mathcal{U}[lo, hi]$
14:        **else**
15:           Define interval $I = [lo, hi] \cap [p - stddev\_scale \cdot w, \; p + stddev\_scale \cdot w]$
16:           Sample $x \sim \mathcal{U}[I]$
17:        **end if**
18:        Add child node with parameter $x$
19:     **end if**
20: **end if**

---

$SampleRolloutList()$ samples a baseline rollout algorithm that is consisted of mutiple iterations of well-working baselines such as scalednewton for sign or scaled Denman-Beavers for matrix square root. If the rollout is coupled iteration but the current state is not coupled, we append the coupling iteration at the front of rollout.

At last, $Backpropogate(s)$ uses Bellman equation to update $V(s)$ in the path from root to $s$ and if $V(s_0)$ is updated, we update bestpath and bestrollout accordingly.

A complete pseudocode for MatRL is as below:

C.3   EXPERIMENTAL ENVIRONMENT

All GPU based experiments were done in NVIDIA RTX A-6000 and CPU based experiments were done in AMD EPYC 7713 64-Core Processor. We repeated the experiments five times and picked the best algorithm, and if the method diverged for five times we ran additional experiments to find a good algorithm.

C.4   HYPERPARAMETERS

Here we detail the hyperparameters in MatRL: this includes basic parameters such as $\alpha$ in progressive widening, list of possible actions for each matrix function, and $RolloutList$ for each matrix function.

We set $C_{pw} = 2, \alpha_{pw} = 0.3, C_{ucb} = 5, E = 5, \epsilon_{tol} = 1e - 6$ and $1e - 11$ for the experiments. The loss function and RolloutList for each matrix function is as below:

Each iteration in ActionList is parametrized to have tunable parameters. A full table denoting how each action is parameterized is as Table 6.

---

**Algorithm 7** MatRL: Monte-Carlo Tree Search for Algorithm Discovery

---

**Input:** $C, \alpha, \epsilon_{\text{tol}}, E, T, ActionList, RolloutList, \mathcal{L}, A \sim \mathcal{D}$, precision, device
Initialize $c, n, t, cp \leftarrow 0, V \leftarrow -\text{INF}, cp[s_{root}] \leftarrow 1$, bestpath, bestrollout $\leftarrow 0$ // Each correspond to number of children, visit count, Transitionable, IsCoupled, and value estimation.
ComputeActionTime(ActionList, A, precision, device)
$U, \Sigma, V^t = SVD(A)$
$(s_{root})_{spectrum} = (\Sigma, I)$
**for** $i = 1$ to $T$ **do**
    $s \leftarrow s_{root}$
    **while** $Expandable(s) ==$ False and $\mathcal{L}(s) \leq \epsilon_{\text{tol}}$ **do**
      // Expandable if $s$ is transitionable and has a child node yet visited, or $c(s) \leq Cn(s)^{\alpha}$
      $s \leftarrow Best_{UCT}(s)$
    **end while**
    $s \leftarrow ExpandNode(s)$ // Here we expand after we look at $cp(s)$
    $r \leftarrow SampleRolloutList()$ // Here we sample from RolloutList, the baselines selected for rollout. If the baseline is coupled but $cp(s) = False$, we attach an additional coupling step at the front
    $s \leftarrow r(s)$
    bestpath, bestrollout $\leftarrow backpropagate(s)$ // Here we use bellman equation. If $V(s_{root})$ was updated, update bestpath and bestrollout
**end for**
**return** bestpath $\oplus$ bestrollout

---

Table 5: Loss function, action list, and rollout list for each matrix function

| Function | Loss Function | ActionList | RolloutList |
|---|---|---|---|
| **Inv** | $\frac{\|AX-I\|_F}{\|A\|_F}$ | `[Inv_NS,` `Inv_Chebyshev]` | `[Inv_NS,` `Inv_Chebyshev]` |
| **Sign** | $\frac{\|X^2-I\|_F}{\|A\|_F}$ | `[Sign_NS,` `Sign_Newton,` `Sign_Quintic,` `Sign_Halley]` | `[Sign_ScaledNS,` `Sign_ScaledNewton,` `Sign_Halley]` |
| **Sqrt** | $\frac{\|X^2-A\|_F}{\|A\|_F}$ | `[Sqrt_DB, Sqrt_NSV,` `Sqrt_Visser,` `Sqrt_VisserCoupled,` `Sqrt_Coupling]` | `[Sqrt_ScaledDB,` `Sqrt_NSV]` |
| **Proot** | $\frac{\|X^3-A\|_F}{\|A\|_F}$ | `[Proot_Newton,` `Proot_Visser,` `Proot_Iannazzo,` `Proot_Coupling]` | `[Proot_Newton,` `Proot_Visser,` `Proot_Iannazzo]` |
| **Rectangular Sign** | $\frac{\|X^T X-I\|_F}{\|A\|_F}$ | `[Cubic, Quintic,` `Septic, Nonic]` | `[Cubic, Quintic,` `Septic, Nonic]` |

Table 6: How actions are parametrized

| Method | Iteration Formula | Parameter Range |
|--------|-------------------|-----------------|
| **Actions for Inverse** | | |
| Newton (Schulz) | $X_{k+1} = a_k X_k - b_k X_k A X_k$ | $a_k, b_k \in [0, 5]$ |
| Chebyshev | $X_{k+1} = a_k X_k - b_k X_k A X_k + c_k X_k A X_k A X_k$ | $a_k, b_k, c_k \in [0, 5]$ |
| **Actions for Sign** | | |
| Newton | $X_{k+1} = \frac{1}{2}(a_k X_k + (a_k X_k)^{-1})$ | $a_k \in [0, 40]$ |
| NewtonSchulz | $X_{k+1} = X_k + a_k(b_k X_k - (b_k X_k)^3)$ | $a_k, b_k \in [0, 5]$ |
| Quintic | $X_{k+1} = a_k X_k + b_k X_k^3 + c_k X_k^5$ | $a_k, b_k, c_k \in [0, 5]$ |
| Halley | $X_{k+1} = X_k(a_k I + b_k X_k^2)(I + c_k X_k^2)^{-1}$ | $a_k, b_k, c_k \in [0, 40]$ |
| **Actions for SquareRoot** | | |
| DenmanBeavers | $\begin{cases} X_{k+1} = \frac{1}{2}(a_k X_k + (b_k Y_k)^{-1}) \\ Y_{k+1} = \frac{1}{2}(b_k Y_k + (a_k X_k)^{-1}) \end{cases}$ | $a_k, b_k \in [0, 50]$ |
| NewtonSchulzVariant | $\begin{cases} X_{k+1} = \frac{1}{2}(a_k X_k - b_k X_k Y_k X_k) \\ Y_{k+1} = \frac{1}{2}(a_k Y_k - b_k Y_k X_k Y_k) \end{cases}$ | $a_k, b_k \in [0, 5]$ |
| Visser | $X_{k+1} = a_k X_k + b_k(A - X_k^2)$ | $a_k, b_k \in [0, 10]$ |
| Visser_Coupled | $\begin{cases} X_{k+1} = a_k X_k + b_k(A - X_k^2) \\ Y_{k+1} = a_k Y_k + b_k(I - X_k Y_k) \end{cases}$ | $a_k, b_k \in [0, 10]$ |
| Coupling | $Y_k = X_k A^{-1}$ | – |
| **Actions for 1/3-th Root** | | |
| Iannazzo | $\begin{cases} X_{k+1} = X_k \left(\frac{a_k I + b_k Y_k}{3}\right) \\ Y_{k+1} = \left(\frac{a_k I + b_k Y_k}{3}\right)^{-3} Y_k \end{cases}$ | $a_k, b_k \in [0, 10]$ |
| Newton | $X_{k+1} = \frac{1}{3}(a_k X_k + b_k X_k A^{-2})$ | $a_k, b_k \in [0, 10]$ |
| Visser | $X_{k+1} = a_k X_k + b_k(A - X_k^3)$ | $a_k, b_k \in [0, 10]$ |
| Coupling | $Y_k = A X_k^{-3}$ | – |
| **Actions for rectangular matrix sign** | | |
| Cubic | $a_k X_k + b_k(a_k X_k - a_k^3 X_k X_k^T X_k)$ | $a_k, b_k \in [0, 5]$ |
| Quintic | $a_k X_k - b_k X_k X_k^T X_k + c_k(X_k X_k^T)^2 X_k$ | $a_k, b_k, c_k \in [0, 10]$ |
| Septic | $a_k X_k - b_k X_k X_k^T X_k + c_k(X_k X_k^T)^2 X_k$ $+(1 - a_k + b_k - c_k)(X_k X_k^T)^3 X_k$ | $a_k, b_k, c_k \in [0, 10]$ |
| Nonic | $a_k X_k - b_k X_k X_k^T X_k + c_k(X_k X_k^T)^2 X_k$ $+(0.9375 - a_k + b_k - c_k)(X_k X_k^T)^3 X_k-$ $0.0625(X_k X_k^T)^4 X_k$ | $a_k, b_k, c_k \in [0, 10]$ |

## C.5 LIST OF DISTRIBUTIONS

The list of distributions we used throughout the experiments are as follows:

1. **Wishart** denotes $A = \frac{X^\top X}{3d} + \epsilon_{\text{stb}} I$ where $X \in \mathbb{R}^{d/4 \times d}$, $X_{ij} \sim \mathcal{N}(0,1)$ i.i.d.. $\epsilon_{\text{stb}} = 1e - 3$ exists for numerical stability.

2. **Uniform** denotes $A = QDQ^T$ where $Q$ is sampled from a Haar distribution and $D$ is a diagonal matrix where its entries are sampled from uniform $[-1, 1]$. We cap the diagonal entries with absolute value $< $ 1e-3 to 1e-3.

3. **Hessian of Quartic** is the indefinite Hessian of a $d$-dimensional quartic $\sum_i z_i^4/4 - z_i^2/4$ evaluated at a random point $z \sim \mathcal{N}(0, \mathbf{I_d})$. We cap the eigenvalues with absolute value ¡ 1e-3 to 1e-3, and normalize with Frobenius norm.

4. **CIFAR-10** is the random input matrix is $\hat{\Sigma} = \frac{1}{n}(X - \mu)^T(X - \mu)$ where $X \in \mathbb{R}^{n \times d}$ is a random batch of $n$ flattened CIFAR-10 images and $\mu$ is the average of the $n$ images. We normalize with the Frobenius norm and add $\epsilon_{stb} I$ for $\epsilon_{stb} = 1e - 3$.

5. **Erdos-Renyi** is the normalized graph Laplacian of a random Erdos-Renyi graph. We set $p = 0.4$ and $d = 5000$ for the experiments.

6. **NanoGPT momentum states** we run the NanoGPT training repository (Jordan et al. (2024a)) and store the momentum state every 125 steps, and split them randomly as train/test data.

# D  LIST OF ALL EXPERIMENTAL RESULTS

## D.1  DIFFERENT MATRIX FUNCTIONS

**Inverse** We learn to compute matrix inverse for two different distributions, Wishart and Uniform. Unfortunately, in our experiments, using Newtonschulz to compute matrix inverse was much slower than directly using torch.linalg.inv. However for uniform distribution, we had a more precise approximation of the inverse in terms of the loss than torch.linalg.inv.

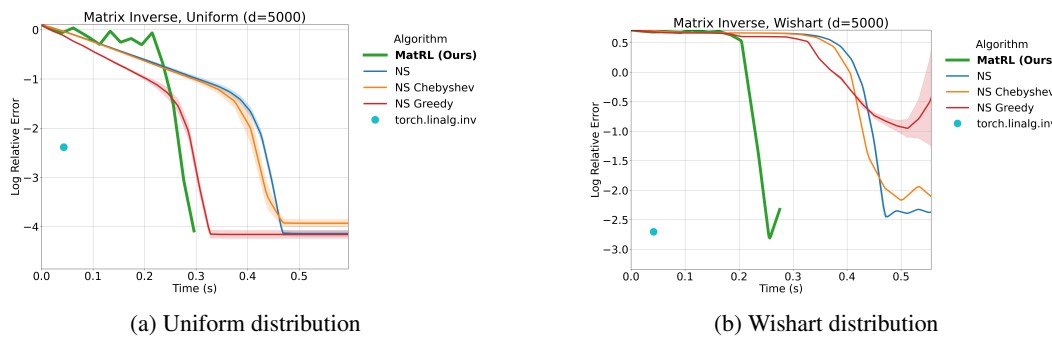

(a) Uniform distribution          (b) Wishart distribution

Figure 5: Computing matrix inverse with NewtonSchulz and variants

**Matrix sign** We learn matrix sign for Quartic Hessian and for matrices with Uniform [-1, 1] diagonal entries. Here $d = 5000$.

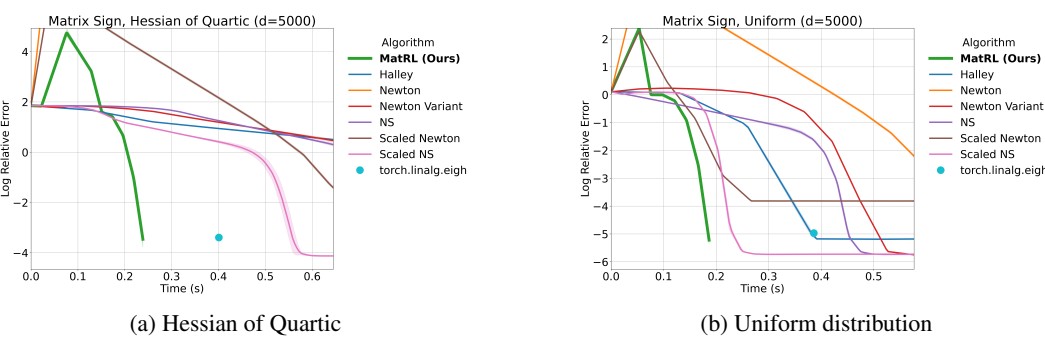

(a) Hessian of Quartic          (b) Uniform distribution

Figure 6: Computing matrix sign with NewtonSchulz and variants

**Matrix sqrt** We learn matrix sqrt for CIFAR-10 and Wishart matrices with $d = 5000$. For CIFAR-10, the matrix is the empirical covariance matrix added by epsilon.

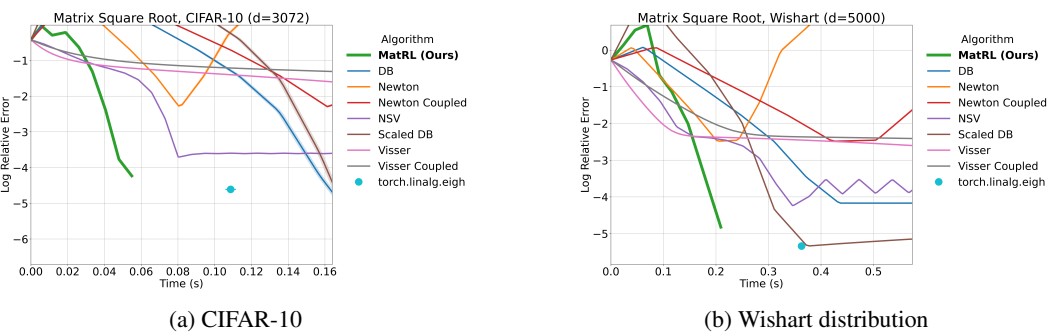

(a) CIFAR-10          (b) Wishart distribution

Figure 7: Computing matrix square root with NewtonSchulz and variants

**Matrix 1/3-root** We learn matrix 1/3-root for Wishart matrices and Erdos-Renyi graph. For Wishart matrices the method is not very effective: torch.linalg.eigh can find matrix 1/3-root with better accuracy with the same amount of time. However, for normalized graph Laplacians of Erdos-Renyi graph, it finds a faster algorithm with almost similar accuracy.

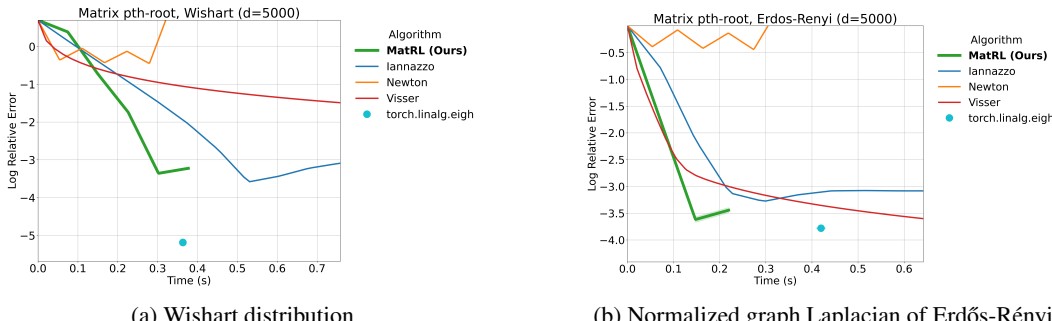

(a) Wishart distribution         (b) Normalized graph Laplacian of Erdős-Rényi

Figure 8: Examples of matrix distributions: structural or spectral views.

### D.2 MATRL ADAPTING: SIZES, PRECISION, COMPUTE

We demonstrate that different $d$, precision (float or double), and compute (GPU/CPU) can lead to different algorithms with matrix sign computation. We show both the loss curve and the found algorithm in each case.

**Different problem sizes** Here we show the results to compute matrix sign on random matrices with spectrum $Unif[-1, 1]$. $d = 1500, 3000, 5000, 10000$. One trend that we see is that for small $d$, we tend to use NewtonSchulz more, whereas for larger $d$ we tend to use Newton step more. It is related with the relative cost between Newton step and Newtonschulz step.

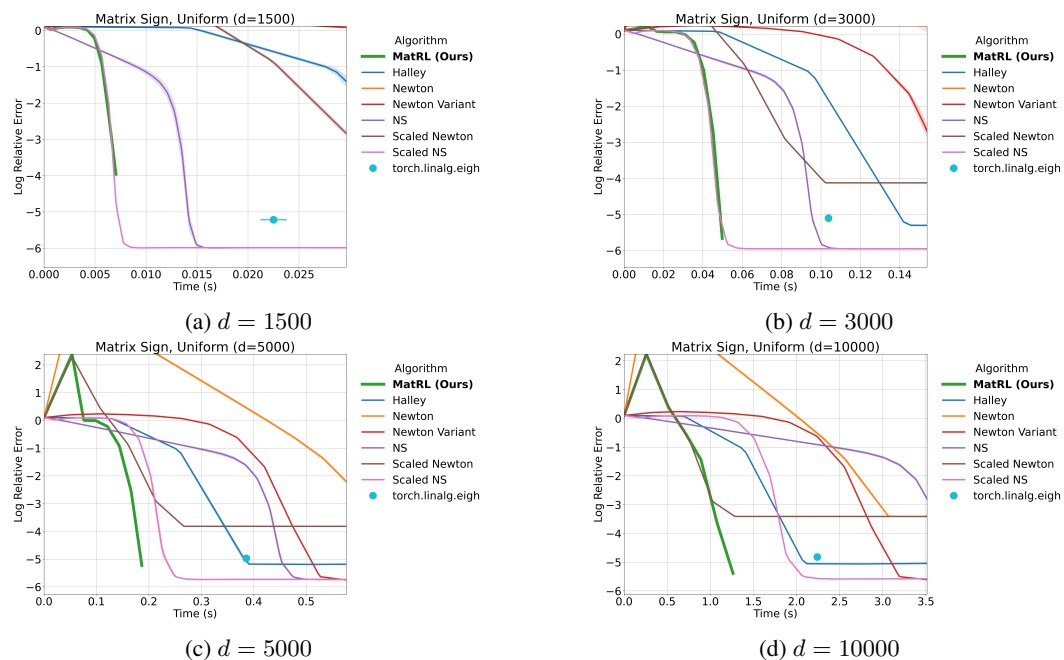

(a) $d = 1500$          (b) $d = 3000$

(c) $d = 5000$          (d) $d = 10000$

Figure 9: Computing matrix sign for different matrix sizes

The found algorithms for $d = 1500, 3000, 5000, 10000$ are as follows.

---

**Algorithm 8** Iterative SIGN for Uniform on GPU with $d = 1500$

---

**Input:** $A$
Initialize $X_0 = A$
Set $a \leftarrow [1.731, 1.729, 1.724, 1.712, 1.682, 1.606, 1.439, 1.190, 1.029, 1.000]$,
// rounded off to three digits
**for** $i = 1$ to $10$ **do**
$\quad X_i = a_{i-1}X_{i-1} + 0.5(a_{i-1}X_{i-1} - (a_{i-1}X_{i-1})^3)$
**end for**
**return** $X_9$

---

**Algorithm 9** Iterative SIGN for Uniform on GPU with $d = 3000$

---

**Input:** $A$
Initialize $X_0 = A$
Set $a \leftarrow [2.179, 1.000], b \leftarrow [1.135, 1.5, 0.5]$
$c \leftarrow [4.308, 1.711, 1.679, 1.599, 1.424, 1.176, 1.025, 1.000]$,
// rounded off to three digits
$X_1 = a_1 X_0 + a_0(a_1 X_0 - (a_1 X_0)^3)$
$X_2 = b_0 X_1 - b_1 X_1^3 + b_2 X_1^5$
**for** $i = 3$ to $10$ **do**
$\quad X_i = 1.5(c_{i-3}X_{i-1}) - 0.5(c_{i-3}X_{i-1})^3$
**end for**

---

**Algorithm 10** Iterative SIGN for Uniform on GPU with $d = 5000$

---

1: **Input:** $A$
2: Initialize $X_0 = A$
3: $a \leftarrow [35.536]$
4: $b \leftarrow [0.094, 1.607, 1.439, 1.191, 1.029, 1]$
5: **for** $i = 1$ to $1$ **do**
6: $\quad X_i = 0.5(a_{i-1}X_{i-1} + (a_{i-1}X_{i-1})^{-1})$
7: **end for**
8: **for** $i = 2$ to $7$ **do**
9: $\quad X_i = 1.5(b_{i-2}X_{i-1}) - 0.5(b_{i-2}X_{i-1})^3$
10: **end for**
11: **return** $X_7$

---

**Algorithm 11** Iterative SIGN for Uniform on GPU with $d = 10000$

---

**Input:** $A$
Initialize $X_0 = A$
Set $a \leftarrow [29.691, 0.243], b \leftarrow [0.616, 1.104, 1.008, 1]$,
// rounded off to three digits
**for** $i = 1$ to $2$ **do**
$\quad X_i = 0.5(a_{i-1}X_{i-1} + (a_{i-1}X_{i-1})^{-1})$
**end for**
**for** $i = 3$ to $6$ **do**
$\quad X_i = b_{i-1}X_{i-1} + 0.5(b_{i-1}X_{i-1} - (b_{i-1}X_{i-1})^3)$
**end for**
**return** $X_6$

---

**Different precision** Here we show the results to compute matrix sign on random matrices with spectrum $Unif[-1, 1]$ for float and double precision. For double precision when $d = 5000$, NewtonSchulz becomes as expensive as Newton step whereas Newton step is more effective - hence we use Newton until the end. For float precision the method finds a mixture of Newton and Newton-Schulz. For double we used $\epsilon_{tol} = 1e - 11$.

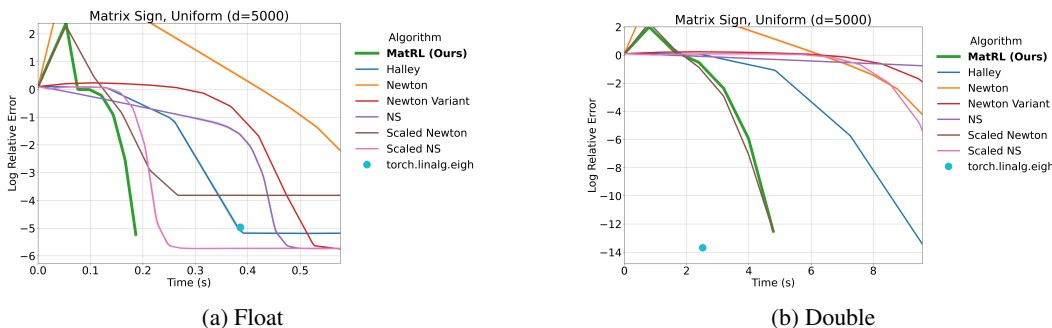

(a) Float

(b) Double

Figure 10: Computing matrix sign for different precision

---

**Algorithm 12** Iterative SIGN for Uniform on GPU with $d = 5000$, DOUBLE

1: **Input:** $A$
2: Initialize $X_0 = A$
3: $a \leftarrow [22.055, 0.244, 0.590, 0.938, 0.998, 1]$
4: **for** $i = 1$ to **6 do**
5:      $X_i = 0.5(a_{i-1}X_{i-1} + (a_{i-1}X_{i-1})^{-1})$
6: **end for**
7: **return** $X_6$

---

**Different compute** We also run MatRL on GPU and on CPU. The difference that occurs here is also similar in vein: on a GPU, Newton step is $\approx$ x2.28 more costly than a Newtonschulz step, whereas on a CPU it is $\approx$ x1.62 more costly. This makes the algorithm found on CPU use Newton step more.

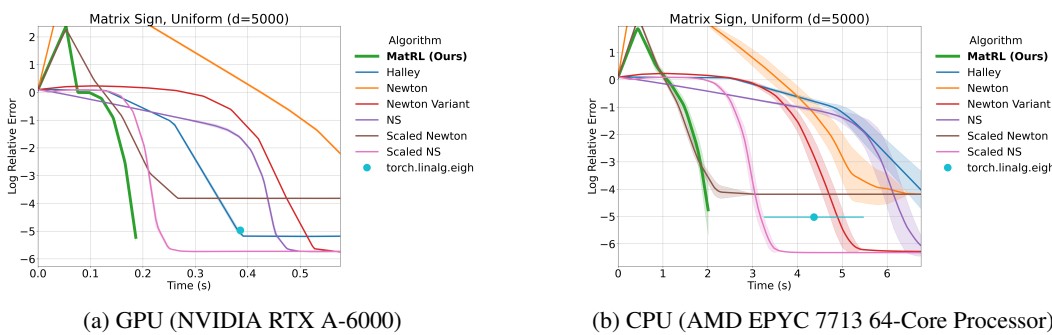

(a) GPU (NVIDIA RTX A-6000)

(b) CPU (AMD EPYC 7713 64-Core Processor)

Figure 11: Computing matrix sign on different machines

---

**Algorithm 13** Iterative SIGN for Uniform on CPU with $d = 5000$

---

**Input:** $A$
Initialize $X_0 = A$
Set $a \leftarrow [19.751, 0.205], b \leftarrow [0.524, 1.161, 1.020, 1.000]$,
// rounded off to three digits
**for** $i = 1$ to 2 **do**
$\quad X_i = 0.5(a_{i-1}X_{i-1} + (a_{i-1}X_{i-1})^{-1})$
**end for**
**for** $i = 3$ to 6 **do**
$\quad X_i = b_{i-3}X_{i-1} + 0.5(b_{i-3}X_{i-1} - (b_{i-3}X_{i-1})^3)$
**end for**
**return** $X_6$

---

### D.3 RECTANGULAR MATRIX SIGN: LEARNED ALGORITHMS

Here we show the matrix sign algorithms found by MatRL that was discussed in Section 5.2. Algorithm 14 and Algorithm 15 are the actual algorithms that were found.

---

**Algorithm 14** MatRL found algorithm, QKVO momentum states

---

**Require:** Initial matrix $A$
**Ensure:** Approximation of $\text{sign}(A)$
1: $X \leftarrow A$
2: coeff_list $\leftarrow$ $\begin{bmatrix} 0.43688178 & 1.6542636 \\ 1.3569574 & 1.1840483 \\ 1.001777 & 0.9366173 \\ 1.9364638 & 1.0367941 \\ 1.1692815 & 1.0191798 \\ 1.658727 & 1.0952345 \\ 2.5182135 & 0.71044815 \\ 0.25106037 & 1.3688384 \\ 1.0987635 & 1.2223818 \\ 1.0320333 & 1.2057912 \\ 1.5708834 & 1.1077545 \\ 0.93263274 & 1.0781184 \\ 2.4632993 & 0.9983091 \\ 0.38100928 & 0.9560407 \\ 0.98132366 & 1.0 \\ 0.5 & 1.0 \\ 0.5 & 1.0 \\ 0.5 & 1.0 \\ 0.5 & 1.0 \end{bmatrix}$
3: **for** each $[a_0, a_1]$ in coeff_list **do**
4: $\quad X \leftarrow a_1 X$
5: $\quad X \leftarrow X + a_0 (X - XX^\top X)$
6: **end for**
7: **return** $X$

---

---

**Algorithm 15** MatRL found algorithm, MLP momentum states

---

**Require:** Initial matrix $A$
**Ensure:** Approximation of $\text{sign}(A)$
1: $X \leftarrow A$
2: $[a_0, a_1] = [1.1972202, 1.2133982]$
3: $X \leftarrow a_1 X$
4: $X \leftarrow X + a_0(X - XX^\top X)$
5: $[a_k, b_k, c_k] = [4.5360823, 8.622313, 7.8011885]$
6: $X \leftarrow a_k X - b_k XX^\top X + c_k(XX^\top)^2 X + (1 - a_k + b_k - c_k)(XX^\top)^3 X$
7: $[a_k, b_k, c_k] = [3.3114827, 5.5237637, 4.457641]$
8: $X \leftarrow a_k X - b_k XX^\top X + c_k(XX^\top)^2 X + (0.9375 - a_k + b_k - c_k)(XX^\top)^3 X - 0.0625(XX^\top)^4 X$
9: $[a_k, b_k, c_k] = [2.8766918, 3.7543924, 2.398172]$
10: $X \leftarrow a_k X - b_k XX^\top X + c_k(XX^\top)^2 X + (1 - a_k + b_k - c_k)(XX^\top)^3 X$
11: $[a_k, b_k, c_k] = [2.25, 2.4375, 1.6875]$
12: $X \leftarrow a_k X - b_k XX^\top X + c_k(XX^\top)^2 X + (0.9375 - a_k + b_k - c_k)(XX^\top)^3 X - 0.0625(XX^\top)^4 X$
13: $[a_k, b_k, c_k] = [2.25, 2.4375, 1.6875]$
14: $X \leftarrow a_k X - b_k XX^\top X + c_k(XX^\top)^2 X + (0.9375 - a_k + b_k - c_k)(XX^\top)^3 X - 0.0625(XX^\top)^4 X$
15: **return** $X$

---

### D.4 ADDITIONAL EXPERIMENTS

Here we present some additional experiments that shows the flexibility of our framework (see Fig. 12). First, we show that MatRL can find good algorithms even though the $\epsilon = 1e-3$ in the Wishart distribution is chosen to be a much smaller value, by running the same experiment on $\epsilon = 1e-5$. Also, we show that MatRL can solve implicit matrix equations such as $X + X^{-1} = A$ when given appropriate base actions. The base actions we give are Newton's method, as well as two fixed-point methods with tunable parameters,

$$X_{k+1} = \alpha A - \beta X_k^{-1}$$

and

$$X_{k+1} = X_k - (\alpha X_k - \beta A)X_k - \gamma I.$$

Fig. 12 shows that MatRL found algorithms can beat existing baselines in these cases, too.

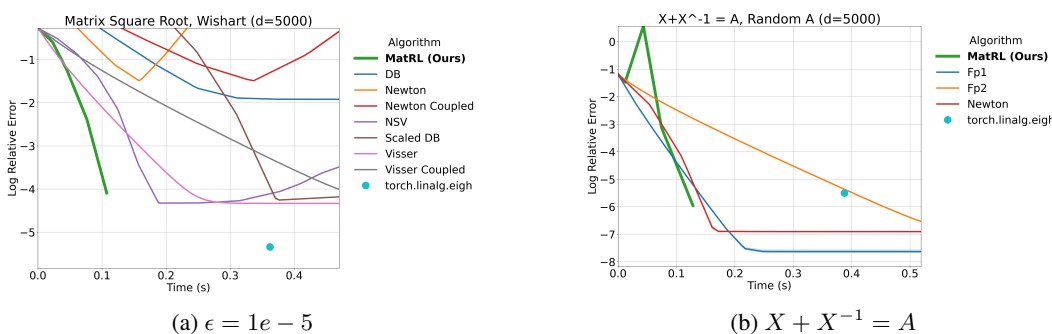

(a) $\epsilon = 1e-5$        (b) $X + X^{-1} = A$

Figure 12: Additional experiments on the flexibility of MatRL

At last, we show that algorithms obtained from the momentum states of NanoGPT small can generalize to the momentum states of NanoGPT-medium. We first see that the limiting spectrum of NanoGPT-small and medium are similar. This becomes a main reason why the learned algorithm can generalize to these matrices.

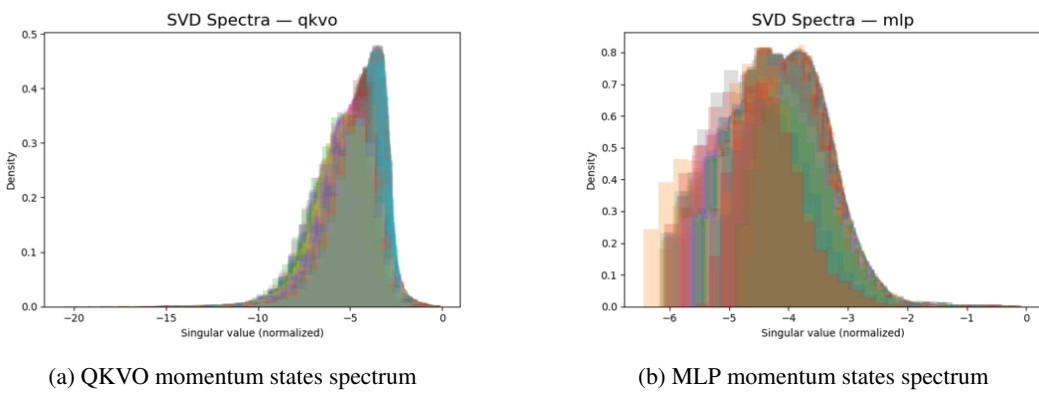

(a) QKVO momentum states spectrum        (b) MLP momentum states spectrum

Figure 13: Spectrum of NanoGPT-medium momentum states

The generalization capability of the learned algorithms in Algorithm 14 and Algorithm 15 can be found in Fig. 14.

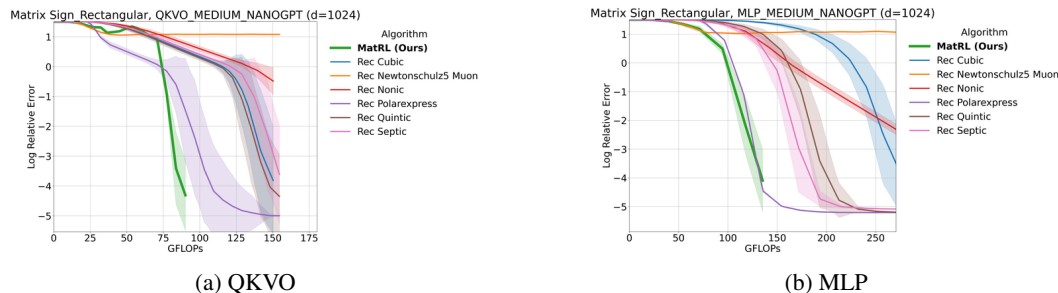

(a) QKVO                    (b) MLP

Figure 14: Generalizing to NanoGPT medium states

## D.5 TIME NEEDED TO FIND EACH ALGORITHM

Here we record the time needed to find each algorithm. For rectangular sign the experiment was repeated 20 times, so the overall time is larger than the other experiments. In general the time to finish one MCTS search was in the order of 5 - 10 minutes, and repeating them added up to less than 3hours in total. Note that once we have an algorithm we can expect it to generalize, and we can reuse it for matrices from the same distribution over and over again.

Table 7: Execution times for various algorithms (in seconds)

| Algorithm | Overall Time (s) |
|---|---|
| inv | 1621.61 |
| inv_wishart | 1445.36 |
| sign | 592.60 |
| sqrt | 528.60 |
| proot | 898.27 |
| sign_d1500 | 374.13 |
| sign_d3000 | 461.22 |
| sign_d10000 | 1073.85 |
| sign_d5000_cpu | 1771.47 |
| sign_d5000_double | 1524.34 |
| sign_hessian | 695.07 |
| sqrt_d10000 | 1121.12 |
| proot_graphLaplacian | 905.72 |
| sqrt_cifar | 727.54 |
| sign_rec_qkvo | 7931.64 |
| sign_rec_mlp | 5004.88 |

