# OpenReview forum: "MatRL: Provably Generalizable Iterative Algorithm Discovery via Monte-Carlo Tree Search"
_ICLR.cc/2026/Conference — Submitted to ICLR 2026_

### Official Review · Reviewer_hhrP · 2025-10-27

**Soundness:** 3
**Presentation:** 3
**Contribution:** 1
**Rating:** 2
**Confidence:** 4

**Summary:**

The paper presents a method for finding matrix algorithms using MC tree search. The MC tree search methodology is relatively standard, what is new is how to apply this framework to finding matrix algorithms, where some design choices were made, an experimental setup is given, and a limiting high probability result for matrix functionals is proven. The paper is clear in exposing its objective, the setup, and it is easy to read. For now though, it's not clear if this approach can be used out of sample.

**Strengths:**

The paper is easy enough to read, and relatively clear. There is some novelty in trying to discover matrix algorithms using MC tree search. I can also image an application. For instance, there is currently a concerted effort to find a matrix sign algorithm that, in 5 iterations (each consisting of one quintic polynomial), in bfloat16 precision, can approximate the matrix sign of the gradient (or momentum buffer).  This is needed to speed up the Muon method, and consequently beat the nanoGPT speed run competition:

https://github.com/KellerJordan/modded-nanogpt/tree/master

But no such clear cut application is given in the paper.

**Weaknesses:**

1. I see a significant issue with train/test distinction and out of distribution performance. Most of the experiments such as Figure 1 and Table 2 are evaluated on the exact same distribution you used to discover your algorithm. In that sense, it is not at all surprising it outperforms the other methods, but this gives little to no indication if the method will perform well out of distribution. If the users objective is to have a method working on one distribution, then you should include the time it takes to find this algorithm for this given distribution. Otherwise, the user could simply take a standard method, and just run it for more iterations, and still save time on having to find this algorithm. Your one example in Figure 2 of fitting an algorithm on $\mathcal{N}(0,1)$ on a Wishhart distribution with the same exact limiting spectrum is still a very limited out of distribution behavior.  Ultimately, for your approach to be useful, you need to show more of the behavior of the algorithms out of distribution.


2. Suppose again that I am user who needs a particular matrix function for a particular application. Your design space and number of hyper parameters is very large, with hyper-parameters
$C,\alpha, \epsilon_{tol}, E, T, D,$ and the choice of actions. How, as a user, can I be confident that it will be easy enough to choose these hyper parameters? Or was there some hyper-parameter tuning for each problem instance? Right now, I would not be confident in using Algorithm 1, and would be more comfortable working in the smaller design space of just one application.


3.  There is a lot of work right now on computing $sign(A)$ because of the Muon method. I know your method is restricted to square symmetric, where as what Muon needs is a method for general rectangular. But you could have still compared your method on square symmetric to several of the competing new methods being developed, such as newtonschulz5/zeropower_via_newtonschulz5 used in Muon.

4. Limited action class, consider polynomials instead. The action class is very limited, with 2 possible updates for Inverse, and 4 for matrix sign. Furthermore, I suppose for the inverses required in the Newton or Halley steps, I suppose (this is not clear in the paper) that you use either a Newton-Schulz or Chebyshev step? Thus ultimately, you are always applying an odd monomial for computing the matrix sign. Why not short-cut this process, and simply restrict your actions to the class of odd monomials? You already consider the Quintic. You could expand upto the ninth degree, and just use that one type of update.

5.   Limited to square symmetric matrices. You are open about this limitation, so I do not really hold it against you. But for now, you rely on the matrix being diagonalizable, and having congruence invariant operators. This excludes applications such as matrix sign applied to the non-square matrices (such as in the muon method you mention). I think you could generalize your setup to rectangular matrices, where you defined congruence using two orthonormal matrices, e.g. $f(UXV^\top) = Uf(X)V^\top$, and you use the singular values as state variables instead of the eigenvalues.

**Questions:**

1. How exactly do you compute the matrix inverse needed in your discovered Algorithm 3 and 4?

2. Do you set the floating point precision in Algorithm 1?

3. For the CIFAR-10 experiment, you report "Log Relative Error". Are you using the same training data or some test/validation data?

** Minor questions***

1. This loss $\mathcal{L}$ in Eq (4) is not defined. At the least, you should say that is some loss function defined by your problem.

2/ On lines 197-233 you used $a\_j$ for the parameters of these step function $f_j$, but on line 245 you introduce this $(k\_1, \ldots, k\_{n\_{j}})$ as the parameters of $f_j$?  Also I assume $k:= (k\_1, \ldots, k\_{n\_{j}})$. Maybe what you meant to say was, that these are all the parameters chosen up to step $n\_{j}$?  I found this a bit confusing.

3. Section 4, you should clarify in the text that by "limiting distribution" you mean "the distribution as the dimension $m$ goes to infinity".

4. Proposition 1: What exactly do you mean by " let .. $f_k^*$ the step $k$ transformation of eigenvalues of the algorithm found by Algorithm 1"? Do you mean, the resulting matrix function after running k steps of your algorithm? This is confusing, since you proposition places no condition on $k$ whatsoever (what about $k=0$?). This proposition is general statement about approximating matrix functions in high probability and high dimensions.  It is not really about your Algorithm 1, but instead a statement about how $P_m$ and $Q_m$ are asymptotically equivalent (or asymptotically indistinguishable) with respect to the functional $L_k.$

---

> ### Author Response · Authors · 2025-11-26
>
> We appreciate the constructive feedback very much and have done our best to address the weaknesses you pointed out, as detailed below.
>
> **1. OOD generalization, lack of clean-cut application**
>
> We have added the modded-nanogpt experiment which shows that MatRL can find good algorithms that can generalize even for out-of-distribution matrices. See general responses 1 for details. In the experiments we would like to point out that the spectrum of momentum states are similar, meaning that our limiting distribution assumption is not only a theoretical assumption but something that we can observe in real world.
>
> **2. Hyperparameter tuning issue of MatRL**
>
> It is true that MatRL has hyperparameters. However, we want to first mention that we did not extensively tune these hyperparameters in all the experiements. Specifically, all the algorithm searches were made by a single set of hyperparameters except for $\epsilon_{tol}$, which depended on the precision and certain matrix functions. Hence we would like to argue that the tuning complexity of MatRL is very light. At last, even though the tuning complexity is high, once you get a good algorithm we can reuse it over and over again, so we believe that it is worth it to use MatRL even in certain cases where tuning may be expensive.
>
> **3. Comparison with Newtonschulz5**
>
> We added two experiments, comparing with newtonschulz5 in Muon and PolarExpress. See general responses 1. We initially did not add newtonschulz5 in Muon because it is a heuristic to approximate matrix sign in Muon and does not converge to the actual matrix sign. PolarExpress was invented to actually find the right matrix sign in a fast and accurate way, and using PolarExpress improves training language models. Nevertheless, we showed both methods and ours and show that MatRL found algorithms can win over existing methods in terms of flopcount.
>
> **4. Limited action class and adding odd polynomials**
>
> First, we admit that inverse has too little action classes. We hope this could be understood as a proof-of-concept rather than a practical algorithm in the inverse case. Thank you for pointing out that we did not specify how inverse was computed in the original manuscript. We explicitly state that we use cholesky + LU solve to compute matrix inverse in the revised manuscript, lines 370-373. In the nanogpt experiment we used more odd polynomials as baselines.
>
> We would like to point out that the power of the framework is that anyone can add any action that anybody wants and come up with a possibly better algorithm. And the action class has no limitations at all - for instance, one can add randomized sketching, deflate, or any other building block that might be useful, and MatRL will find a good algorithm. Hence we believe that the proposed framework actually has no limitations in the action class, and we believe this is a major benefit of the framework.
>
> **5. Limitation to square symmetric matrices.**
>
> Thank you for the valueable suggestion. We added how the framework could be extended to rectangular matrices in lines 292-298, and did modded-nanogpt experiments (see general response 1).
>
> **6. Do you set the floating point precision in Algorithm 1?**
>
> Yes we do. Specifying precision and compute is now shown in figure 1, and also in the manuscript, line 273.
>
> **7.  This loss in Eq (4) is not defined.**
>
> We mentioned that $\mathcal{L}$ is a loss function to be defined later in the revised manuscript.
>
> **8.  Notations on $k_1, \cdots k_{n_j}$**
>
>  We fixed it to $a_{j1}, \cdots a_{jn_j}$.
>
> **9. The meaning of limiting distribution**
>
> We added the meaning of the limiting distribution in line 322.
>
> **10.  Meaning of $f_k^{\ast}$ in proposition 1**
>
> Yes we are assuming $X_k = Qf^{\ast}_k(\Sigma)Q^{T}$. It is true that this proposition is a general statement about approximating matrix functions in high dimensions with high probability. You can think that an application of this proposition shows that MatRL learned algorithms will perform well to matrices with similar spectrum. We specified the meaning of $f_k^{\ast}$ in the proposition.

---

> ### Comment · Reviewer_hhrP · 2025-11-26
> **Appreciated, but too big a change now to properly review**
>
> Dear Authors,
>
> I do appreciate your answer, but this too big a change for me to properly review this during this phase. I think with your new changes, this paper may be ready for resubmission. But I can't now commit the time needed to re-review this. Though I will raise my score from 2 to 4 because you have addressed some of my questions.
>
> Your new experiments on Nano-GPT are appreciated, but raise many questions. How did you split train and test for these momentum matrices? What was the cost of the training phase? Does it generalize across model sizes for GPT models ? What exactly was the odd polynomials you discovered, and how far are they from these baseline polynomial methods? Since this is now the only out of distribution experiment you have, the practicality of your method rests on these results.
>
> For 4. in computing the inverses using  cholesky+ LU solve, how are you running this on GPUs? Both methods have typically bad access patterns on GPU.

---

> ### Author Response · Authors · 2025-11-26
>
> Dear reviewer hhrP,
>
> Thank you for raising your score and raising some important questions on out-of-distribution generalization. For an easier review, we give a short concise list of changes that we made to the manuscript. Most of the changes that we made were modifications in the presentation, and the NanoGPT experiment was the main new experiment that we did. We would like to emphasize that these are not completely new experiments that are unrelated with the previous paper, but are applications and extensions of our existing theorems / experiments to a real-world example.
>
> [NanoGPT experiment]
>
> - We show that we can find better matrix sign computation methods by applying MatRL on momentum states of NanoGPT repository.
> - We show the approximate limiting distribution behavior of momentum states for NanoGPT training.
> - We discuss how MatRL can leverage the expanded algorithm search space by not sticking to only quintic iterations.
>
> [Presentation]
>
> - We add figure 1 to clearly explain the process of MatRL.
> - We fixed lines 263-298 for clarity.
>
> Here we answer to the questions that you raised:
>
> 1. We store the momentum states every 125 steps for all layers and parameters, and split train/test dataset by randomly choosing whether the matrix should be in training dataset or the test dataset, for each matrix.
>
> 2. The time needed for algorithm discovery is noted in Table 7 of the appendix. Each MCTS run needs the order of minutes (5~10 minutes), and multiple runs to find a good algorithm can increase the overall time.
>
> 3. We haven't yet done experiments about the generalization across different model sizes, though we believe results such as [1] implies the universality of spectrum density in neural networks in some sense. We will add the experimental results on how the method generalizes to models with different sizes.
>
> 4. In Appendix D.3., we report the learned algorithms for qkvo and mlp momentum states. For early stages of the algorithm the parameters deviate, but in the later stages it converges to the original parameters. We can describe a similar phenomenon happening in PolarExpress, where at the initial stages the parameters deviate much from [1.875, -1.25, 0.375] (which are the parameters for quintic approximation for matrix sign), but it converges to those parameters at later stages.
>
> 5. We share the submodule _safe_inverse that we used to compute matrix inverse.
>
> ```
> def _safe_inverse(M: torch.Tensor) -> torch.Tensor:
>     try:
>         L = torch.linalg.cholesky(M)
>         return torch.cholesky_inverse(L)
>     except RuntimeError:
>         LU, piv = torch.linalg.lu_factor(M)
>         n = M.shape[-1]
>         I = torch.eye(n, dtype=M.dtype, device=M.device)
>         if M.ndim > 2:  # batched case → repeat identity along batch dims
>             I = I.repeat(*M.shape[:-2], 1, 1)
>         return torch.linalg.lu_solve(LU, piv, I)
> ```
>
> References:
>
> [1] Hodgkinson, Liam, Zhichao Wang, and Michael W. Mahoney. "Models of Heavy-Tailed Mechanistic Universality." arXiv preprint arXiv:2506.03470 (2025).

---

### Official Review · Reviewer_i9wr · 2025-10-31

**Soundness:** 3
**Presentation:** 3
**Contribution:** 3
**Rating:** 8
**Confidence:** 2

**Summary:**

This paper introduces MatRL, a reinforcement-learning-based framework that automatically discovers iterative algorithms for computing matrix functions. The approach frames algorithm design as a sequential decision-making process, using Monte Carlo Tree Search to select and tune hybrid iterations. A key theoretical contribution is that the learned algorithms are shown to generalize to large matrices drawn from the same distributions, and with the same limiting eigenvalue spectrum. The authors demonstrate the method’s effectiveness on tasks such as matrix inversion, square roots, and sign functions in their experiments. There are several open directions and follow-up works, including extending these to rectangular and sparse matrices.

**Strengths:**

1. It is a novel framework and a new connection with reinforcement learning
2. The results generalize to large matrices and there are provable guarantees.
3. The experiments suggest that the method works better than non hybrid ones.
4. A key idea they exploit is that different methods work better on different parameter regimes such as condition number etc. The framework allows to use the ideal method in the required parameter regime without explicitly determining which regime you are in. I think this can help with the design of optimization algorithms in general.

**Weaknesses:**

1. Does not extend to rectangular matrices.
2. This seems like a new concept, so maybe some more basic explanations/background might help.

**Questions:**

1. Are there any works on this in the past?
2. Is there a high-level reason why these don't work for rectangular matrices?
3.    How large are the search and training times?

---

> ### Author Response · Authors · 2025-11-26
>
> We appreciate the positive response and constructive feedback from the reviewer, and have done our best to address the weaknesses you pointed out.
>
> **1.Weakness on rectangular matrices**
>
> See general responses 1. We showed how we can extend the framework to rectangular matrices in the revised manuscript.
>
> **2.Adding more background information**
>
> We appreciate the feedback that we need additional explanations/background explanation. We have background explanation on MCTS, iterative algorithm discovery for matrix functions, and automated algorithm discovery. We explicitly added that recent algorithm discovery frameworks define novel search space, and our work falls in the framework. If you suggest topics that need more basic explanation, we would appreciate it and add them to the paper.
>
> **3.Any related work in the past?**
>
> There are quite a few recent automated algorithm discovery papers, but we are the first to discuss the topic in iterative linear algebra algorithm space. See general responses 3 for further details.
>
> **4.High level reason why it doesn't work for rectangular matrices**
>
> We initially restricted ourselves to square matrices because we thought it is easy to model theoretically and define matrix functions clearly. Following the original definition in [1], for general matrices the matrix function is defined as :
>     given a matrix $A$ and its Jordan canonical form $A = PJP^{-1}$, $f(A) = Pf(J)P^{-1}$ [1] where for each Jordan block $J_k$, $f(J_k)$ is defined as
>     $$
>     f(J_k) :=
>     \\begin{bmatrix}
>     f(\\lambda_k) & f'(\\lambda_k) & \\cdots & \\frac{f^{(m_k-1)}(\\lambda_k)}{(m_k-1)!} \\\\
>     0 & f(\\lambda_k) & \\cdots & \\frac{f^{(m_k-2)}(\\lambda_k)}{(m_k-2)!} \\\\
>     \\vdots & \\ddots & \\ddots & \\vdots \\\\
>     0 & \\cdots & 0 & f(\\lambda_k)
>     \\end{bmatrix}.
>     $$
>     So we thought it could be overly complicated to discuss general matrix functions for general matrices. However, if we define matrix functions in a slightly different way, we can extend the result to rectangular matrices too. See general responses 1, and lines 292-298 of the revised manuscript.
>
> **5. How large are the search and training times?**
>
>  In section D.5 we added a table that shows the time needed to find each algorithm. Each MCTS run needs the order of minutes (5~10 minutes), and multiple runs to find a good algorithm can increase the overall time. Note that once we find an algorithm we can reuse it as much as we can, and the time to find an algorithm is not a bottleneck as long as we can find a good one.
>
> References:
>
> [1] Higham, Nicholas J. Functions of matrices: theory and computation. Society for Industrial and Applied
> Mathematics, 2008.

---

### Official Review · Reviewer_r7Uu · 2025-11-01

**Soundness:** 2
**Presentation:** 1
**Contribution:** 2
**Rating:** 4
**Confidence:** 3

**Summary:**

This paper presents a reinforcement learning framework for learning matrix algorithms. This task is cast as a reinforcement learning problem where states represent intermediate steps of the computation, actions represent (parametrized) transformations, and the reward captures computational cost. The approach is demonstrated on several algorithmic tasks such as matrix sign and matrix square root, and appears to provide computational benefits compared to exist direct (eigenvalue decomposition) and iterative methods.

While I see some merit in this direction of research, I have a hard time understanding what the paper does and how experiments are conducted. Therefore, I believe the paper needs substantial improvements in the presentation clarity before being published.

**Strengths:**

* The paper considers the automated discovery of matrix algorithms, which has the potential to improve computational workloads by finding more efficient algorithms
* Numerical experiments suggest that computational improvements can be achieved compared to existing baselines / state-of-the-art algorithms

**Weaknesses:**

* The paper is, overall, hard to follow. It is easy to get lost in the various notations, and some terminologies are used without a clear definition
* The proof of several key theoretical results is deferred to the appendix, which makes it hard to validate the findings
* The experiments appear to be executed on a single matrix (see questions below), and the paper does not appear to discuss performance variability
* Proposition 1 appears to be an existential result based what seem to be relatively strong assumptions (identical spectrum distribution), which states (as far as I understand), that

**Questions:**

* In the proposed RL scheme, how is $\mathcal{L}$ captured? I did not see it mentioned in the reward, nor does it appear in Algorithm 1
* The presented RL setup considers a value function $V(s, a)$ that depends on both state and action. However, for the obtained algorithm to be applicable in practice, the policy should not depend on the state, only of the iteration number (this is because the state information comprises the matrix' spectrum, hence, one needs an eigenvalue decomposition to know the full state). Can the authors please explain how this is tackled in their approach?
* In proposition 4.1, I do not understand the relation between $m$ and $k$, can the authors please provide more context and explanations?
* Is the MCTS training phase executed on a _single_ matrix?
* Similarly, I have a hard time understanding whether the results reported in Section 5 are evaluated on a single matrix, or are averages over multiple matrices. If it's the former, results on a single sample may not be indicative of general performance. If the latter, some metric of performance variability would be useful
* In the SQRT experiments, I would expect that a cholesky factorization would be faster to compute that eigenvalue decomposition

---

> ### Author Response · Authors · 2025-11-26
>
> We appreciate the constructive feedback and have done our best to address the weaknesses you pointed out, as detailed below.
>
> **1. Issues with the presentation**
>
> We admit that our initial manuscript was very hard to follow. We tried to improve presentation a lot by adding an abstract schematic of the algorithm as well as rewriting the not-clear paragraphs. See general responses 4 for details.
>
> Moreover, we had to put many of the results in the appendix because of the page limit. We tried our best to point at references to the appendix at places where needed. Also, we moved certain parts that we thought is important to be in the main paper from the appendix, such as experimental settings. If you recommend certain parts of the paper that should not be in the appendix and should be in the main paper, we would appreciate it and accomodate the suggestion.
>
> **2. The experiments appear to be executed on a single matrix, issues with the evaluation of the algorithm**
>
> How we evaluated the algorithm has now made clearer. See general responses 2.
>
> **3. Weakness on Proposition 1**
>
> The feedback was not completed, so we would appreciate it a lot if the reviewer could clarify their claim. One thing that we would like to note is that the convergence to the same spectrum is motivated from Marchenko-Pastur, where the matrices have the same limiting spectrum under rather weak assumptions, such as i.i.d. mean 0 variance 1. Also, our revised experiments on the spectrum of NanoGPT momentum states(figure 3) show that the limiting spectrum phenomena is not a theoretic assumption, but happens in real world.
>
> **4. Question on how $\mathcal{L}$ is captured**
>
> It is one of the termination conditions, see equation (4). Admittedly, it was not clearly presented before in the manuscript. In figure 1, we explicitly show that the loss function acts as a termination criterion.
>
> **5. How to remove the dependency of $s$ in $V$**
>
> Thank you for asking an important question! Yes, it is indeed true that if we learn $V(s,a)$ in some way and have to compute the next action every time we iterate, it can be even more computationally expensive than just running the vanilla algorithm a few more times. And it would not make any sense to compute the spectrum then plug it into $V$, because once we finish the SVD we can simply use it to compute matrix functions.
>
> The point here is that the initial spectrum is similar for all matrices from the same distribution. Hence, once we find "a single" algorithm that works for $A$ sampled from $\mathcal{D}$, it can generalize to other matrices with the same distribution. Here is where the generalization guarantee becomes useful: because we have a guarantee we can use a found algorithm to others, for the original matrix $A$ we can find an algorithm that actually depended on the spectrum on $A$, then reuse it on multiple different matrices from the same distribution.
>
> To summarize, the assumption on the limiting spectrum erases the dependency on $s$.
>
> **6.Relation between $m$ and $k$**
>
> $m$ is the size of the matrix and $k$ is the step. The theorem says that for sufficiently large matrices (so that the spectrum is similar to the limiting spectrum), the $k$-th step losses are also similar, hence generalization happened. We added explanations in lines 333-335.
>
> **7. In SQRT, Cholesky factorization would be faster**
>
> We would like to clarify that the matrix square root we are interested in is $X^2 = A$, whereas Cholesky decomposition solves $XX^{T} = A$. Matrix square can be defined in various different ways, and there are definitions that declare $X$ satisfying $XX^{T} = A$ is the matrix square root. As our definition for matrix square is different, we cannot directly use the Cholesky decomposition as matrix square root. If there are other methods that can solve $X^2 = A$ faster using Cholesky decomposition, we would appreciate it and compare it with our method.

---

### Official Review · Reviewer_udHw · 2025-11-01

**Soundness:** 3
**Presentation:** 2
**Contribution:** 2
**Rating:** 6
**Confidence:** 3

**Summary:**

The paper proposes a reinforcement learning algorithm based on Monte Carlo Tree Search to discover faster algorithms for computing functions of large matrices. The authors demonstrate the approach on several example functions, including the matrix sign and square root. The proposed RL method appears effective at identifying efficient algorithms.

**Strengths:**

-	The RL algorithm managed to identify algorithms with better performance on the examples presented in the paper.

**Weaknesses:**

The training is applicable only when the state is reduced to the spectrum of the matrix and to functions within the Congruence Invariant Diagonal Preserving framework. This likely limits the general applicability of the resulting algorithms.

The results presented in Section 4 on generalization appear rather weak, as they essentially concern only matrices with asymptotically similar spectra. The title therefore seems somewhat optimistic. A deeper investigation of the algorithms’ generalization properties would have been valuable.

There is no apparent novelty in the reinforcement learning algorithm itself.

**Questions:**

Line 208: the loss function ${\cal L}$ is not specified below … it is done much later on examples.
Typo in eq (4): $n_i$ should be $n_{t_i}$.
The notation used 242-245 is not appropriate, before $k$ referred to the iteration number, here it refers to the vector $a_{j$ -- may be a better notation would be $a_j^{(k)}$ the vector of parameters used in the $k$-th iteration when using the function $f_j$.
I find the discussion Line 250-259 unclear, while the message seems trivial – because the authors restrict themselves to a Congruence Invariant Diagonal Preserving setting.
The same holds for discussion starting Line 260. The set of possible actions is indded huge. What does at stage $1,ldots,n_j$, only parameters $k1,\ldots,k_{n_j−1}$ are chosen: what do you mean? Don’t you select $k_{n_j}$ too?

---

> ### Author Response · Authors · 2025-11-26
>
> We appreciate the constructive feedback and have done our best to address the weaknesses you pointed out, as detailed below.
>
> **1. The training is applicable only when the state is reduced to the spectrum of the matrix and to functions within the Congruence Invariant Diagonal Preserving framework.**
>
> First, we would like to argue that many important functions such as polynomials, matrix sign, root, exponential, logarithm, etc can be understood as congruence invariant diagonal preserving. Also, we believe our experiments suggest that our framework, even though it is limited, is applicable to various matrix functions and baseline algorithms. At last in the revision we extended some of our results to rectangular matrices (see general responses 1). We believe it would be an interesting extension to apply this to functions that do not satisfy the property.
>
> **2. On Proposition 1 and the generalization guarantee**
>
> We appreciate the feedback that a more thorough investigation on generalization would be valueable. We first would like to mention that the asymptotically similar spectrum is a universal property in random matrices, and it happens in rather weak assumptions, e.g. when the entries of X are sampled i.i.d. mean 0, variance 1. Such random matrix theory results exist in real-world matrices like weight/gradient matrices of neural networks [1], and we added figure 3 in the revised manuscript to emphasize that the limiting spectrum assumption happens in real world, not just a theoretic assumption.
>
> **3. There is no apparent novelty in the reinforcement learning algorithm itself.**
>
> Admittedly, we do not propose a novel reinforcement learning algorithm that can be generally used. However, we believe we have scientific novelty that we design a space of algorithms and propose a search strategy for the space. See general responses 3 for details.
>
> **4. Line 208: the loss function ${\cal L}$ is not specified below … it is done much later on examples.**
>
> We specified that $\mathcal{L}$ can be chosen by the user and we only specify the class here in the revised version.
>
> **5. Notation issue on lines 242-245**
>
> We changed the $k_j$s in the environment to $a_{j1}, ... a_{jn_j}$.
>
> **6. Issues in the presentation of lines 250-259 and 260**
>
> We clarified the paragraphs in the revised version. Specifically, we mean in lines 250-259 that by using spectrums we do not have to use matrix-matrix multiplications in the MCTS run, which makes the search faster ($O(n^3)$ when using real matrix multiplications to $O(n)$ when using spectrums.) Using decoupled actions is beneficial because we can avoid exponential exploring (exponential in terms of $n_j$). For a big picture see figure 1 of the revised manuscript.
>
> We fixed the typos that the reviewer caught. Thank you so much for reading the paper thoroughly and finding the typos.
>
> References:
>
> [1] Hodgkinson, Liam, Zhichao Wang, and Michael W. Mahoney. "Models of Heavy-Tailed Mechanistic Universality." arXiv preprint arXiv:2506.03470 (2025).

---

### Official Review · Reviewer_GaWh · 2025-11-05

**Soundness:** 3
**Presentation:** 2
**Contribution:** 2
**Rating:** 4
**Confidence:** 3

**Summary:**

This paper proposes a method to compute certain functions of matrices efficiently. The idea is to use reinforcement learning to tune the parameters within certain pre-existing iterative methods for computing these matrices, and to decide when to jump from one iterative method to another.

The paper seems to produce nice empirical results, showing that the tuned methods are indeed very fast ways of computing certain matrix functions, taking advantage of the actual runtime observed on your specific hardware to tune these runtimes. It's neat.

The method should apply to any spectral function $f(A)$ so long as it is efficient to compute some form of residual between the current iterative and $f(A)$. They use the square root and matrix sign functions in this paper.

The paper is fundamentally empirical, and most of the effort in the paper is put into characterizing how the space of iterative methods can be fit into a reinforcement learning framework. Theory is given, but doesn't exactly give any qualitative rates beyond the existence of an epsilon-delta guarantee.

**Strengths:**

The paper is interesting.

I'm pretty borderline on it because I'm not super clear how much "new science" there is in it; though if the paper is considered to be sufficiently novel then I don't really have any other concerns about its publication. I'm a bit of an outsider to RL as a community, so it's hard for me to judge how much this paper is really cool RL research (as opposed to like good engineering on top of standard RL tools).

The motivation is compelling. We want to compute spectral functions efficiently. Lots of existing methods exist. It's really annoying to figure out which iterative method you should be using right now. Machine learning seems like a good toolkit for answering this question.

The experiments are interesting and suggest that a large speedup is often possible. Would be nice to see this on a larger variety of functions that what's currently considered -- namely for a fancier function like log(A) or e^A or something like that.

The example algorithms generated, algos 2 3 and 4 on page 8 are pretty great payoffs.

Overall, I'm pretty cool with this paper. The intro is well written and the idea is simple but seems likely helpful.

**Weaknesses:**

The paper is confusing in places. The experiments are still weaker than I'd want them to be, in terms of validating a proposed mathematical algorithm on a wide variety of matrices.

I don't really understand all the twists and turns in the precise engineering of the titular MatRL algorithm [page 6]. The pseudocode is a bit too abstract, and the importance of all the elements feels a bit vague to me. I wouldn't be surprised this feels vague to me because I'm not as familiar with the RL side of thing though. _I don't really understand where the matrix A is even used in this algorithm_

The paper needs a bit reflowing as well. It's left very vague for a long time what kinds of degrees of freedom we have when designing iterative methods. For instance, Newton iteration shows that the procedure $X \leftarrow \frac12(X + X^{-1} A)$ converges to $X = A^{1/2}$. It is unclear though what parameters we might optimize here; can we replace $\frac12$ with something else? The nature of this question isn't really made clear enough early enough in the paper; a running example would be helpful here.

There's this paragraph on page 5, "Spectrum as state variables", which seems to suggest storing the eigenvalues of the matrices used in the iterative method as part of the state of the RL algorithm to reduce memory cost. This really confuses me because computing the eigendecomposition is slower than using an iterative method to compute f(A). Further, the RL algorithm needs to actually run primitives like matrix-matrix products on real matrices in memory to see 1) the actual runtime of these matrix-matrix products and 2) the numerical stability of the method examined. So I'm pretty thoroughly confused on this point.

Real-world evaluation of the proposed method is a bit lacking. There are huge datasets of matrices easily findable (eg suitesparse) where we can easily find many viable real-world input matrices. It'd be great to see a much larger number of matrices undergo the treatment of this proposed RL algorithm and the histogram of "ratio of MatRL runtime vs torch.linalg.eigh runtime". This would be needed for practitioners to seriously consider using the proposed method for finetuning their runtimes.

The precision achieved by all methods, including dense eigenvalue computation, is strangely low, capping out at 4 digits of precision. Perhaps this is an artifact of using 32 bit floats and using an error metric that squares our matrices (thereby square the condition numbers involved)?

In their experiments, they use a pretty darn large regularization of $10^{-3} I$ to ensure numerical stability. I'm not clear why this is so large, as this generally limits the accuracy of methods to not exceed about 3 digits of accuracy. I'm used to something much closer to machine epsilon, though maybe this is close to the square root of the machine epsilon? Either way, seems large to me.

The experiments have no confidence intervals, showing that they do or don't act consistently. Since their method involves some randomness in exploring their RL framework (to my understanding at least...), it seem like there should be some confidence intervals.

**Questions:**

Is definition 1 really meaningfully different than the definition of a spectral function?

## Typos & Recommended edits

Feel free to ignore anything in this list without further discussion

1. [throughout] use \citep where needed, so that citations are much easier to read around
2. [53] add units to the 2.9 and 1.7 numbers
3. [71] say "method" instead of "solution"
4. [72] "a desiderata of" to "desiderate from"
5. [76] "which iteration to use" isn't clear language here. Maybe "iterative method" is better?
6. [81] remove "algorithm"
7. [101] I disagree with this sentence. Other methods exists for accessing matrix functions. Krylov method. Eigendecompositions. Shrink the scope, perhaps.
8. [192] "where D is a distribution over random matrices defined on"
9. [216] Add "diagonal preserving"
10. [218] remove one of the two math expressions; I think the former one.
11. [222] comma after "is"
12. [222] "to find" instead of "finding"
13. [259] typo on spectrum
14. [Decoupled Actions on page 5] I don't get what's going on here. It's not clear to me how the continuous search space is searched, and it's not clear why decoupled actions resolves issues around continuous search spaces
15. [Decoupled Actions on page 5] I don't really follow the algorithmic idea here. The paragraph is just confusingly phrased to me.
16. [278] add "the" after "depending on"
17. [295] UCB should be UCT. Or the rest of the paper needs to say UCB.
18. [321] I don't understand why you need the probabilities around the convergence statements. Can't you just say that mu(X) and mu(Y) both weakly converge to mu*?
19. [326] Given that the set [a, b] is closed, do we actually needs the epsilons? Can't we actually just replace this with either [a, b] or (a, b)?
20. [351] Fix the quotes before "Hessian"
21. [Figs 1 and 2] what does the shaded region mean? How many trials was this run over?
22. [throughout] How did it take for MatRL to run to find methods like Algos 2, 3, and 4

---

> ### Author Response · Authors · 2025-11-26
>
> We appreciate the very detailed constructive feedback and have done our best to address the weaknesses you pointed out, as detailed below.
>
> **1. Comment on "how much new science" is in this paper.**
>
> See general responses 3.
>
> **2.  Extension to fancier functions such as matrix exponential or logarithm**
>
> Thank you very much for the suggestion. We did not do experiments on matrix exponential or log, because a general method to compute those matrix functions is using a pade' approximation of the function rather than using iterative methods [1]. Specifically, in [1], it is written that for matrix exponentials the scaling and squaring method and the schur form are the most prominent methods, and for matrix logarithms they use Pade' approximations. Nevertheless, we thought it would be cool to show that MatRL can find better algorithms for more complex functions. Hence we added an experiment where MatRL finds a better algorithm to find a value of the implicit matrix function
>     $$
>     X + X^{-1} = A
>     $$
>     for Wishart random matrices, mixing Newton's method and two fixed-point methods. See section D.4. for details.
>
> **3. Twist and turns of MatRL and clarity issues**
>
> We admit that the pseudocode for MatRL and the initial explanation was a bit vague. We replaced it with figure 1 for a clear, intuitive explanation of what is happening, and we moved the pseudocode to the appendix. We also rewrote the lines 263-298 for clarity.
>
> **4. Another clarity issue: e.g. what are the parameters that we can optimize for each iteration? It is unclear now when we read the paper.**
>
> We added a running example in figure 1 using Visser iterations. Also, we explicitly point to the relevant tables in the appendix in the main paper now (see lines 315-317), which are Tables 5 and 6 in appendix C.
>
> **5. Weakness on the elaboration of section "spectrum as state variables"**
>
> We admit that the initial explanation was not clear on how we benefit from using spectrum as states. It is true that we need to run matrix-matrix products on real matrices at least once. What we do is for each possible action(or the function), we benchmark the function once at the initial stage of the search, and use the benchmarked time as a penalty when each action is used. This is now clearly stated in figure 1. Now using eigenvalues as state becomes an efficient implementation choice because if we use real matrix-matrix multiplication in each MCTS run we need $O(n^3)$ complexity, whereas if we use only spectrum interactions the complexity is $O(n)$.
>
> **6. Weakness of lacking real-world evaluation**
>
> We admit that the initial version of the manuscript had restricted real-world experiments. We initially looked into suitesparse, but we thought it might not be a good benchmark because matrix functions of sparse matrices are not sparse in general, and though we believe it is very interesting to extend the current work to sparse matrices, our current version does not exploit sparsity. We added the experiment on the momentum state of NanoGPT instead, which not only demonstrates that our method can find better algorithms in real world, but also that our assumption on the spectrum is happening in real world. See general responses 1 for further details.
>
> **7. Weakness: The precision is low for all experiments. Is it an artifact of floating-point precision?**
>
> Yes, it is an artifact of using 32bit float. We have much better precision when the precision is set to double (see figure 10 in the appendix).
>
> **8. Weakness: Large regularization of $\epsilon=1e-3$**
>
> The reason we used $\epsilon$ in the paper was to ensure that the random matrix is not singular. We show that MatRL indeed works for smaller $\epsilon$ in the revised manuscript, section D.4.
>
> **9. Weakness: No confidence intervals**
>
>  In the experiments where the training and test matrices were sampled from the exact same distribution, the runtime for all matrices were very similar so that even though we showed confidence intervals it was hard to see. For nanogpt experiments it is more visible, see Figure 4 - (a), (b) of the revised manuscript. Also, for all experiments we explicitly mentioned that we added the shaded band corresponding to one standard deviation.
>
> **10. Question: Is definition 1 simply a spectral function?**
>
> Definition 1 is more or less a spectral function, but we are only interested in the case when the arguments have the same left/right singular vectors (since it makes sense to only define functions that are only related to the spectrum in these settings).
>
> References:
>
> [1] Higham, Nicholas J. Functions of matrices: theory and computation. Society for Industrial and Applied Mathematics, 2008.

---

> ### Author Response · Authors · 2025-11-26
>
> Also, we appreciate the reviewer for suggesting various recommendations for edits. We agreed on most, and fixed many weird/confusing expressions due to the valueable suggestions. We would like to highlight some important feedback that the reviewer gave:
>
> **On recommendation 14, 15**
>  We rewrote the paragraph for better explanation. The core idea is that when we understand the $n_j+1$ tuple as a whole, we have to do exponentially many explorations which is inefficient. We decouple the exploration of parameters so that the search can concentrate on the more probable set of parameters.
>
> **On recommendation 18**
>
> An event happening deterministically and an event happening by probability 1 is different. For instance, when we sample a real number from [0,1], the probability that we sample a rational number is 0, but it can happen in infinitely many different ways. As $\mu_m(X), \mu_m(Y)$ are random variables, a proper way to state is using probability. The assumption is motivated from Marchenko-Pastur, which states that $\mu_m$ converges to $\mu^{*}$ with probability 1 under certain assumptions on the distribution.
>
> **On recommendation 19**
>
> We cannot. We will show a running example that shows the statements
>     $$
>     \lim_{m \rightarrow \infty} \mathbb{P}(S_m \subseteq [a-\epsilon, b+\epsilon]) = 1
>     $$
>     for all $\epsilon > 0$ and
>     $$
>     \lim_{m \rightarrow \infty} \mathbb{P}(S_m \subseteq [a, b]) = 1
>     $$
>     are different.
>
> Suppose $S_m = [a-1/m, b+1/m]$ with probability $1-1/m$ and $[a, b]$ with probability $1/m$. $S_m$ satisfies the previous condition because for given $\epsilon > 0$, if $m$ is sufficiently large, $\mathbb{P}(S_m \subseteq [a-\epsilon, b+\epsilon]) = 1$ and its limit is also 1. However, $\mathbb{P}(S_m \subseteq [a, b]) = 1/m$, which goes to 0 as $m \rightarrow \infty$. Note that $\lim_{m \rightarrow \infty} \mathbb{P}(S_m \subseteq [a, b]) = 1$ implies $\lim_{m \rightarrow \infty} \mathbb{P}(S_m \subseteq [a-\epsilon, b+\epsilon]) = 1$, hence we are proving a strictly stronger theorem.
>
> **On recommendation 22**
>
> In section D.5 we added a table that shows the time needed to find each algorithm. Each MCTS run needs the order of minutes (5~10 minutes), and multiple runs to find a good algorithm can increase the overall time.

---

### Author Response · Authors · 2025-11-26
**[General Response]**

We express our sincere gratitude to the reviewers. Due to the detailed comments, our paper could improve a lot. Here we address some most common concerns that the reviewers had.

**1. New experiment on NanoGPT momentum states**

Many of the reviewers had concerns about the limited scope of the paper, including the restriction to symmetric square matrices, limited empirical results, and the limitation of the generalization theorem which only works for matrices with the same limiting distribution. In the revised version of the manuscript we added applying MatRL directly to the optimizer states to the NanoGPT training repository in section 5.2. Specifically, we stored the momentum states (of which matrix sign is applied to) of the modded nanogpt repository (https://github.com/KellerJordan/modded-nanogpt) and trained MatRL on cubic, quintic, septic, and nonic polynomial actions. We can see that MatRL can find better algorithms to compute matrix sign in terms of flopcount compared to existing baselines, including the original newtonschulz used in Muon and PolarExpress[1]. The major gain using MatRL is that we don’t stick to quintic polynomials as these methods do, and possibly use other iterations that lead to better algorithms. Moreover, we show that the spectrum of these momentum states are similar in a sense, meaning that our “same limiting eigenvalue distribution” assumption happens in real world. We hope the experiment can relieve concerns about
- the method being restricted to square symmetric matrices
- limited empirical results
- How well the algorithm generalizes to OOD distributions and if the generalization theorem is realistic/too weak

**2. On train/test splits**

A few reviewers mentioned about training/testing specifics - e.g. was the algorithm trained over a single matrix? was it tested on a single matrix? was the training and test data separated? Admittedly, it was unclear in the paper how we trained/tested the algorithm. To clarify, we always trained on multiple matrices, and also tested on many independent samples. In the case of known distributions, each training sample and test matrix is a brand new iid sample from the distribution. In the case of real-world matrix datasets, we explicitly separated train/test datasets. Also, all experiments except the
NanoGPT experiment was tested on 10 randomly sampled test matrices, and the NanoGPT experiment was tested on 100 randomly sampled test matrices.

We added a paragraph in the revised manuscript explaining how the experiments were made in lines 362-369.

**3. Scientific significance**

A few reviewers mentioned that the weakness of this work is that we do not propose new RL algorithms. We admit that we do not propose a new RL algorithm that can be generically used. The scientific significance we want to propose is that we design a new environment, or a search space, where we can make the possibly infinite space of algorithms tractable, and then propose an efficient search strategy that can find a good algorithm in the environment. Several recent advancements in algorithm discovery [2], [3], [4] do not really concentrate on proposing a new RL framework or method, but define an environment or search space that is tractible then use existing methods in a clever way to find better algorithm. Our work is the first to deploy this idea into finding iterative numerical linear algebra algorithms, and we believe we have scientific significance in this sense.

We added a brief explanation in lines 82-85.

**4.  Presentation Issues**

Many of the reviewers were concerned of the presentation issues, especially in section 3. We replaced the abstract algorithm with a figure that can be more understandable in an intuitive way, and rewrote most of lines 263-298. We fixed the typos that the reviewers kindly found and expressions that reviewers recommended to fix for clarity.

References:

[1] Amsel, Noah, et al. ”The polar express: Optimal matrix sign methods and their application to the muon algorithm.” arXiv preprint arXiv:2505.16932 (2025).
[2] Alhussein Fawzi, Giacomo De Palma, Ankit Goyal, Gary Becigneul, Mohammad Barekatain, Sam Bond-Taylor, et al. Discovering faster matrix multiplication algorithms with reinforcement learning. Nature, 610(7930):47–53, 2022. doi: 10.1038/s41586-022-05172-4.
[3] Daniel J Mankowitz, Andrea Michi, Anton Zhernov, Marco Gelmi, Marco Selvi, Cosmin Paduraru, Edouard Leurent, Shariq Iqbal, Jean-Baptiste Lespiau, Alex Ahern, et al. Faster sorting algorithms discovered using deep reinforcement learning. Nature, 618(7964):257–263, 2023.
[4] Yann Collet, Nick Terrell, W Felix Handte, Danielle Rozenblit, Victor Zhang, Kevin Zhang, Yaelle Goldschlag, Jennifer Lee, Daniel Riegel, Stan Angelov, et al. Openzl: A graph-based model for compression. arXiv preprint arXiv:2510.03203, 2025.

---

### Author Response · Authors · 2025-12-03
**[Final Comment]**

We would first like to thank the area chair and the reviewers for putting much effort in revising and improving the paper. For an easier evaluation of the revised manuscript, we show a summary of the concerns each reviewer had, and how we mitigated the issues.

**Reviewer GaWh** was pretty positive about the paper, but had concerns related to both insufficient experiments, clarity. By insufficient experiments the reviewer wanted additional matrix functions and lacking sufficient real-world experiments. We added new experiments on NanoGPT momentum states (see general responses 1) and implicit function computation (see Appendix D.3.) to show that MatRL can find better algorithms for both real-world matrices and more complex matrices. The reviewer also had issues with clarity, especially on how MatRL works (section 3). We added Figure 1 instead of the abstract pseudocode to make the method more accessible, and revised the writing in section 3.  At last, we clarified many concerns/issues raised by the reviewer, mostly on the writing.

**Reviewer udHw** mainly discussed on the generalization guarantee that we have in the paper, as well as that there is no novelty in the RL algorithm itself. For generalization guarantee, we now have out-of-distribution generalization experiments where the spectrum are not identical but similar, by showing that the learned sign function generalize well in NanoGPT momentum states. We also have experiments where the learned algorithm generalize to different sizes of NanoGPT experiments. We admitted that we do not have an apparant general strategy that can be used for reinforcement learning - but we proposed our scientific significance in general response 3.

**Reviewer r7Uu** was especially not very happy about the presentation of the paper, and how the training/test matrices were distinguished. We improved the presentation, especially in section 3, and specified how train/test matrices were specified in section 5. Specifically, we mentioned that we test with different randomly sampled matrices and aggregated them.

**Reviewer i9wr** was quite positive about the paper, and raised questions about novelty, as well as weakness of square symmetric matrices. We answered with the NanoGPT experiment and discussions on scientific significance.

**Reviewer hhrP** raised various concerns on out-of-distribution generalization, lack of clean-cut application, lack of baselines (e.g. NewtonSchulz in Muon), and limitation to square symmetric matrices. We mitigated the issue with NanoGPT momentum states experiment. Specifically, we showed that the spectrum of QKVO and MLP momentum states have similar limiting distribution, as well as the limiting distribution generalizes with different NanoGPT experiment sizes. We also added PolarExpress and NewtonSchulz-5 in Muon as baselines, and added odd polynomials as actions. We also clarified the hyperparameter tuning cost for MatRL, how we computed matrix inverse, and reported the time of each algorithm search.

At last, we would like to propose that our paper is the first to find iterative numerical linear algebra algorithms by RL techniques. The main novelty comes from defining a new environment/search space that is tractable then using existing methods in a clever way to find a better algorithm that is useful. With the emerging artificial intelligence technology, our paper is a starting point of exploration and optimization of iterative linear algebra algorithms with AI.

---

### Meta-Review · Area_Chair_khtS · 2026-01-07

**Summary:**

For this work, Reviewers found its framework novel and its results interesting. Among five initial reviews, two of them are positive. However, Reviewers also have concerns on the presentation of this paper, scientific significance, and experiments. The author rebuttal and the paper revision addressed some of these concerns, for example, I agree with the authors that not every paper needs to propose a new RL algorithm, but the presentation remains a big concern. Specifically, the amount of the paper revision work goes beyond a short round of conference paper review, so the paper could benefit a lot from a careful and detailed revision. Also, the support for accept from reviewers is a bit weak. Overall, this is a borderline paper. Given all five reviews, after reading the author rebuttal and considering the potential of raising scores, I still find that this work is not good enough to be published at ICLR right now, therefore, I recommend reject.

**Reviewer Concerns:**

Reviewers have concerns on the presentation of this paper, scientific significance (for example the new RL algorithm discussion), and experiments. The author rebuttal and the paper revision addressed some of these concerns, but the presentation remains a big concern.

**Reviewer Scores:**

Only Reviewer hhrP had some discussion with the authors and raised the score from 2 to 4 after reading the author rebuttal. For two reviewers who gave positive ratings, Reviewer udHw and Reviewer i9wr, I don't think they would raise their ratings since their initial support is not very strong. For Reviewer GaWh and Reviewer r7Uu, they might raise their scores, so the final scores would be very diverse, leaning towards a borderline rating.

---

### Decision · Program_Chairs · 2026-01-26

Reject